# Unbiased Classification Through Bias-Contrastive and Bias-Balanced Learning

**Youngkyu Hong**[†]
Naver AI Lab
youngkyu.hong@navercorp.com

**Eunho Yang**
KAIST, AITRICS
eunhoy@kaist.ac.kr

## Abstract

Datasets for training machine learning models tend to be biased unless the data is collected with complete care. In such a biased dataset, models are susceptible to making predictions based on the biased features of the data. The biased model fails to generalize to the case where correlations between biases and targets are shifted. To mitigate this, we propose Bias-Contrastive (BiasCon) loss based on the contrastive learning framework, which effectively leverages the knowledge of bias labels. We further suggest Bias-Balanced (BiasBal) regression which trains the classification model toward the data distribution with balanced target-bias correlation. Furthermore, we propose Soft Bias-Contrastive (SoftCon) loss which handles the dataset without bias labels by softening the pair assignment of the BiasCon loss based on the distance in the feature space of the bias-capturing model. Our experiments show that our proposed methods significantly improve previous debiasing methods in various realistic datasets.

## 1 Introduction

Machine learning models have achieved extremely high performance in a variety of tasks and domains such as computer vision and natural language processing [15, 7, 39]. Recently, however, many concerns have arisen that such evaluation does not reflect the real-world performance of the model when it is deployed [35, 41]. Among others, failing especially due to the biases existing in the dataset can lead to serious societal side effects such as prejudice or racism beyond simple algorithm failure [45, 34, 5]. If bias features are highly correlated with the object class in the dataset, models tend to use the bias as a cue for the prediction as they are easier to learn but enough to achieve high accuracy even though they are not actually related to the target class [3, 33].

The failure of learning due to the bias existing in the dataset appears in various fields and tasks. To name a few, in image classification task, [18, 30] discovered that state-of-the-art CNNs have texture biases. Visual question answering (VQA) model is also known to be susceptible to biases as it only uses the word occurrence in the question to generate answer [1, 11]. For example, if most of the images of banana in the train sets are yellow, the model directly answers the question "What is the color of the banana?" as "Yellow", without looking at the image of green banana. Surveillance models, which can cause serious societal problems if misclassified, tend to give biased predictions toward sensitive attributes such as race [34].

Considering the importance of the problem, many approaches have been proposed to mitigate such biases in the training dataset. Depending on the presence or absence of bias information, the approaches can be divided into two categories. When the bias label is available, [28, 43] add the bias prediction branch for the unbiased prediction. [40, 10] directly regularize the feature embedding to be indistinguishable across the bias classes. When the bias label is unavailable, [11, 3, 8, 20] design

---

[†]This work was done as a student at KAIST.

35th Conference on Neural Information Processing Systems (NeurIPS 2021).

an auxiliary bias-capturing models that learn only bias features and train the main model to learn orthogonal features from those models.

In this paper, we propose two powerful debiasing approaches that can be complementarily applied when the bias label is available. We first propose **Bias-Con**trastive (BiasCon) loss that extends the approaches that directly regularize the feature space by adapting the recent advances of contrastive learning in representation learning. The BiasCon loss utilizes the power of contrastive learning to promote pulling the same target class but different bias class samples closer in the feature space. We further propose the **Bias-Bal**anced (BiasBal) loss that suppresses the utilization of bias features by optimizing toward the data distribution where the target-bias correlation is balanced. Each loss achieves state-of-the-art debiasing performance and shows even higher performance when they are jointly used as they give orthogonal debiasing effects.

To extend to more realistic cases where the bias label is unavailable, we use the observation that the feature space of the bias-capturing model can be used in estimating whether a pair of samples have the same bias features. We thus propose **Soft** Bias-**Con**trastive (SoftCon) loss, which is the BiasCon loss weighted with the cosine distance between samples in the feature space of the bias-capturing model.

We conduct experiments to evaluate the debiasing performance of the proposed methods. For the case where the bias label is available, we evaluate the methods on CelebA [31] and UTKFace [46], which have biases toward sensitive attributes such as gender or race. For the case where the bias label is unavailable, we use ImageNet [36] and ImageNet-A [23] to assess whether the bias of our model has been removed. Our method improves the unbiased accuracy of previous methods by a large margin across all datasets in both with and without bias labels.

Our contributions can be stated as follows:

- We propose a powerful debiasing method, the BiasCon loss, that effectively adapts recent advancements of contrastive learning and the BiasBal loss that further enhances debiasing performance by optimizing the model toward distribution with uniform target-bias correlation.

- We introduce the SoftCon loss, the extension of the BiasCon loss to the case where the bias label is unavailable, that utilizes the feature space of the bias-capturing model.

- We show that our losses successfully improve the debiasing performance with a large gap in various real-world datasets for both bias label available/unavailable cases.

## 2    Related work

Bias existing in the data and the vulnerability of machine learning algorithms to such bias has been recently studied as an important problem in various domains and tasks. [35, 41] revealed that various training sets have regularity conditions (e.g., objects should not be occluded) that are unlikely to hold in practical settings, and machine learning models trained on such data fail to generalize in the absence of such conditions. More specifically, [23, 44, 30] showed that state-of-the-art object recognition models are biased toward backgrounds or textures associated with the object class. [38] explored that overparametrized models are prone to the spurious correlations and can have worse test errors on minority groups. Natural language processing models are also susceptible to dataset biases. Question Answering (QA) [19, 25, 13, 12] and Visual QA [1, 11] systems make predictions using shortcuts such as the occurrence of specific words in the question rather than the context. Even large language models tend to give biased prediction toward certain gender or race [10]. Meanwhile, the severity of such biased prediction and fairness issues of deployed models are thoroughly investigated in various domains [45, 34, 5]. This brings the importance and necessity of the debiasing method, and recent research has begun to proceed in two main directions depending on the presence or absence of bias labels.

**Debiasing when bias label is available**    An important line of work assumed the situation where the bias factors are categorical attributes (e.g., gender, race) and the meta-information about the bias label of each training data is available. [43] suggested to use per-bias classifier head to equalize the effect of bias features. [45, 32, 14] proposed fairness metrics such as demographic parity (DP) or equalized odds (EO) that measure the biasedness of the model and devised algorithms that can

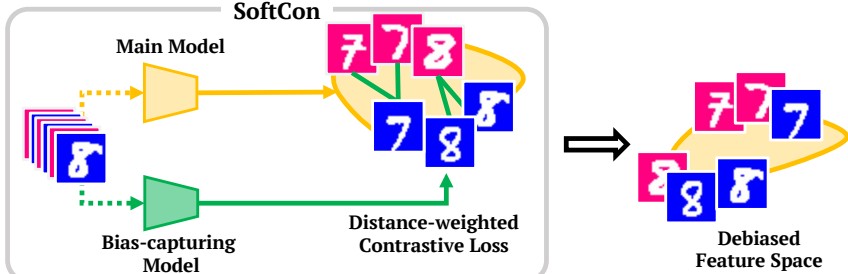

Figure 1: Visualized explanation of our SoftCon loss. As samples with similar bias features are close in the feature space of the bias-capturing model, we design SoftCon loss that does the distance-weighted contrastive learning to pull a pair with the same target class but with different bias features.

induce the fair prediction. [28] extended the idea of the domain generalization [2, 17] by adding a bias prediction branch to the model and train the model with an additional regularization term that induces the learned features to be invariant to the bias features. In the risk minimization perspective, [4, 16, 37] proposed distributionally robust optimization that aims to optimize toward the worst-case group distribution shift. [40] devised a regularization term with a triplet loss formulation to minimize the entanglement of bias features. For the pretrained language model, [10] appended additional filtering layer and corrects the sentence embedding by maximizing mutual information. We propose a novel regularization term that work upon the success of the contrastive learning framework on self-supervised learning [9, 21, 27] and leverages the bias label in assigning positive pair samples.

**Debiasing when the bias label is unavailable**    It is more realistic to assume that we know the form of biases that have to be removed, but no bias label is available, nor the type of bias is not limited to categorical data (e.g., texture bias of CNNs). Still, there are approaches that mitigate biases in such situations. The first line of research is to utilize the bias-capturing model [11, 3, 8, 20]. The bias-capturing model is an auxiliary model designed to learn bias features. For example, texture biases can be captured by CNNs with small receptive fields [3]. In these approaches, the model is promoted to learn independent features from the bias-capturing models, thus learning debiased features. [33] further relaxed the assumption about the prior knowledge about bias based on the observation that the bias features are learned before the target features. Specific to CNNs for the image classification task, [18, 30] removed texture bias by randomizing the texture of the training data. Our method also utilizes a bias-capturing model, but we propose a more effective way to use the model by considering the distance between training samples in the feature space of the model.

## 3   Method

This section begins by explaining our problem setup. Then, we propose two debiasing objectives, **Bias-Con**trastive (BiasCon) loss and **Bias-Bal**ancing (BiasBal) loss, that are effective in situations where the bias is categorical and the bias label is available. We further propose **Soft** Bias-**Con**trastive (SoftCon) loss which extends the BiasCon loss to more general and challenging situations where the bias is not limited to categorical data and the bias label is not available.

### 3.1   Problem definition

To formulate our problem, we define a target $Y$ and a bias $B$ that both affects the generation of data $X$. In such data, we assume that $X$ can be divided into a signal feature $X_s$ explaining the target $Y$ and a bias feature $X_b$ from $B$. For example, in the data for hair color classification task that is biased toward race (that is, $Y$=hair and $B$=race), $X_s$ includes brown hair feature, and $X_b$ includes white skin feature. Now, the bias in the dataset can be understood as a high correlation between the target class and the bias feature, i.e., the prior $p(Y|B)$ is imbalanced. Then the model may ignore the true signal $X_s$ and instead use $X_b$ in prediction and output $p(Y|X_b)$ as it can still give high accuracy. In the hair color example above, if most white people have blonde hair, the model may output blonde every time it sees the white skin color. Therefore, the biased model fails on the test data distribution with the shifted correlation between $Y$ and $B$, i.e., $P_{train}(Y, B) \neq P_{test}(Y, B)$.

Our objective is to train the model to make a prediction using the true signal $X_s$ only; that is, the model takes the whole feature $X$ and outputs $p(Y|X_s, X_b) = p(Y|X_s)$ conditionally independent on $X_b$, so that the model makes indiscriminative predictions across the biases.

## 3.2 Bias-contrastive loss

We first consider the situation where the bias label is categorical and known for all training samples (e.g., gender or race information of each training sample is available). Here we note the fact that a *biased* model tends to learn features from $B$ and make predictions using those features, hence the samples within the same bias class are close to each other in the feature space. Thus, we devise a loss to explicitly incentivise the model to pull the same target class but *different* bias class sample pairs closer than the other pairs.

Toward this, we borrow and revise the main idea of [27] where supervised contrastive (SupCon) loss pulls the same class samples closer using the contrastive loss and achieves better performance than training solely with the cross-entropy loss. Specifically, we turn the SupCon loss into an effective loss for debiasing, named **Bias**-**Con**trastive (BiasCon) loss, that pulls samples with the same class but different bias closer while pushing the other pairs.

Following the formulation of the SupCon loss, we first construct $2N$ size "multiviewed batch" by applying a pair of randomized augmentation (e.g., random crop, random flip) on $N$ data samples in a single batch: $\{(x_i, y_i, b_i)\}_{i=1}^{2N}$. We then apply the same SupCon loss but to different *positive* pairs (pairs of same target class but different bias class) and formulate the BiasCon loss as follows:

$$L_{\text{BiasCon}} = -\frac{1}{2N} \sum_{i \in I_{mv}} \frac{1}{|J(i)|} \sum_{j \in J(i)} \log \frac{\exp(z_i \cdot z_j / \tau)}{\sum_{a \in I_{mv} \setminus \{i\}} \exp(z_i \cdot z_a / \tau)} \tag{1}$$

where $I_{mv} := \{1, ..., 2N\}$ is the index set of multiviewed batch, $J(i) := \{j \in I_{mv} : y_j = y_i, b_j \neq b_i\}$ is the index set of positive samples paired with $i$-th sample, $z_i = f(x_i)/\|f(x_i)\|$ is a normalized feature of $i$-th sample extracted from $f$, the network up to the penultimate layer, and $\tau$ is a temperature hyperparameter.

The final objective is a combination of our BiasCon loss and the standard cross-entropy (CE) loss as follows:

$$L_{\text{CE}} = -\frac{1}{N} \sum_{i \in I} \log h(x_i)[y_i], \tag{2}$$

$$L = \alpha \cdot L_{\text{CE}} + L_{\text{BiasCon}} \tag{3}$$

where $h(x)[y] = p(y|x)$ is the model prediction for class $y$, $I := \{1, ..., N\}$ is the index set of the original batch and $\alpha$ is a weight hyperparameter. Originally, [27] used the SupCon loss solely for the representation learning up to the penultimate layer and additionally trained the fully-connected (FC) layer upon the frozen representation with the CE loss. However in our case, samples with the same target and bias class are considered as a negative pairs. Therefore, it is more effective to jointly train with CE loss as it can give signals to incorporate all the same class samples. Thus, unlike [27], we propose to use the BiasCon loss as a regularizer, as in (3). In the Appendix D.2, we empirically show that the joint training with CE loss achieves better performance.

Another specific issue here that contrasts with the SupCon loss is that when $p(b|y)$ is highly imbalanced, a single batch may not contain enough positive pairs because most of the same class samples in the batch will have the same bias features. This may result in unstable optimization. To handle this issue, for training set $\mathcal{D} = \{x_i\}$, we separately sample the mini-batch for the BiasCon loss with a modified sampling frequency $Q(i)$ of $i$-th sample. We design $Q(i)$ to oversample $x_i$ with low $p(b_i|y_i)$ so that more positive pairs are available in the batch:

$$Q(i) \propto \begin{cases} \frac{1}{p(b_i|y_i)} & \text{if } \frac{1}{p(b_i|y_i)} < \gamma \cdot \min_{y,b} \frac{1}{p(b|y)} \\ \gamma \cdot \min_{y,b} \frac{1}{p(b|y)} & \text{o.w.} \end{cases}. \tag{4}$$

The mini-batch of size $N$ is then sampled as $\{x^{(j)} : x^{(j)} = x_i$ where $i \sim Q(i)$ for $j = 1, \ldots, N\}$. Note that this formulation clips the largest possible unnormalized $Q(i)$ to be $\gamma \cdot \min Q(i)$ so that every training data is sampled with some chance. We pre-calculate $p(b|y)$ using the $Y$ and $B$ labels in the training set before the training.

It is also instructive to note that EnD [40] has the similar motivation of entangling the features of different bias class samples. However, EnD uses the triplet loss which has a limitation that it penalties the same class samples even after the model successfully learned the target features. Experimental results in Section 4 show the huge performance gap between ours and EnD across all tasks. More detailed comparison between two methods is available in Appendix A.2.

### 3.3 Bias-balanced regression

Highly imbalanced $p(y|b)$ inhibits the learning of features (i.e., $x_s$) that can correctly classify the minor data under the same bias label. Here, we propose a novel method called bias-balanced regression, which debiases the model by optimizing the model toward the data distribution with the uniform correlation between $Y$ and $B$. Specifically, let $P_{train}(X, Y, B)$ be the data distribution of the training set and let $P_u$ be the unbiased test data distribution with the uniform correlation between Y and B. Here, the test data can be understood as the shifted distribution from $P_{train}$:

$$P_u(X, Y, B) := P_{train}(X|Y, B)P_u(Y, B), \tag{5}$$

$$P_u(Y, B) := P_u(Y|B)P_{train}(B) = \frac{1}{C} \cdot P_{train}(B), \tag{6}$$

where $C$ is the number of target classes. Biased model fails to generalize to $P_u$ as the correlation between $Y$ and $B$ in $P_{train}$ is not applicable in $P_u$. To handle this, we aim to use $P_u(X, Y, B)$ in training to optimize the model performance on the data distribution without correlation between $Y$ and $B$. However, due to the bias in the training data, we cannot directly access the samples of $P_u(X, Y, B)$ and need to achieve this goal from biased $P_{train}$.

Toward this, consider the model prediction $p(y|x, b)$ and the corresponding Bayesian interpretation:

$$p(y|x, b) = \frac{p(x|y, b)p(y|b)}{p(x|b)}. \tag{7}$$

As can be seen from (7), the bias of training data causes a distributional shift of $p(y|b)$ in the unbiased test phase, resulting in discrepancy of predictive distributions $p(y|x, b)$ during training and test. To correct this discrepancy, inspired by [26], we derive the theorem below:

**Theorem 1.** *(Bias-balanced Regression) Assume the multinomial logistic regression with the model $h(x)[y] = \frac{e^{\eta_y}}{\sum_{y'} e^{\eta_{y'}}}$ where $\eta_y$ is the logit for class $y$, and the training data distribution $P_{train}(X, Y, B)$. Let $P_u$ be the data distribution shifted from $P_{train}$ to have the uniform correlation between Y and B. If the model $h$ estimates the conditional probability over $P_u$, i.e., $h(x)[y] = p_u(y|x)$, then the estimate of the conditional probability over $P_{train}$ is:*

$$p_{train}(y|x) = \frac{\exp\left(\eta_y + \log p_{train}(y|b)\right)}{\sum_{y'} \exp\left(\eta_{y'} + \log p_{train}(y'|b)\right)}, \tag{8}$$

*and the cross-entropy loss risk associated with $h$ over $P_{train}$ is $\mathbb{E}_{P_{train}}[L(h(x), y, b)]$, where $L(h(x), y, b)$ is defined as follows:*

$$L(h(x), y, b) := -\log \frac{\exp\left(\eta_y + \log p_{train}(y|b)\right)}{\sum_{y'} \exp\left(\eta_{y'} + \log p_{train}(y'|b)\right)}. \tag{9}$$

We name the loss function defined in (9) as **Bias-Bal**ancing (BiasBal) loss:

$$L_{BiasBal} := \frac{1}{N} \sum_{i \in I} L(h(x_i), y_i, b_i), \tag{10}$$

where $I$ inherits the definition in (2). The BiasBal loss can be used as a replacement for the CE loss, hence it can be seamlessly combined with other existing debiasing methods based on CE loss. In the Appendix D.3, we empirically show that the BiasBal loss further improves the performance of existing methods. Especially, it is well suited with the BiasCon loss, resulting in superior performance in all tasks (Section 4.1).

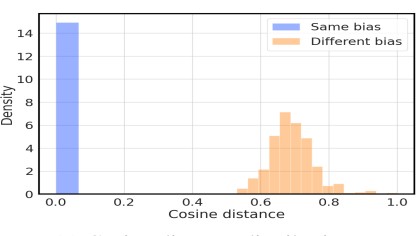
(a) Cosine distance distribution

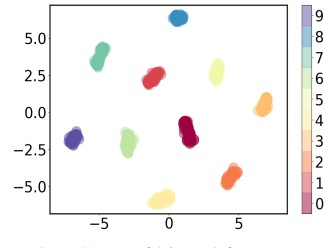
(b) t-SNE of biased features

Figure 2: (a) Distribution of cosine distances between sample pairs of same / different bias classes in the feature space of the bias-capturing model trained on BiasedMNIST [3]. (b) t-SNE [42] analysis of biased features shows the clear separation between bias classes.

### 3.4 Extension to the unknown bias label case

We extend our algorithm to more general settings that do not assume the type of biases to be categorical or the availability of bias labels. However, following [3], we still assume that we have prior knowledge about the form of the bias and we can design a bias-capturing model that inherently make prediction only with $x_b$, i.e., model predicts $p(y|x_b)$. For example, a CNN model with a small receptive field can only learn local features, thus it only allows to use texture biases to make predictions. (e.g., predict zebra only with stripe features.) We use a pre-trained bias-capturing model as an auxiliary model for debiasing the main model.

To gain information about the bias of the dataset through the lens of the bias-capturing model, we focus on the feature space of the bias-capturing model. Intuitively, when a pair of samples are close to each other in the feature space of the bias-capturing model, it is likely that two samples have similar bias features in order for the bias-capturing model to work. To check this intuition, we consider a BiasedMNIST [3] dataset, an MNIST [29] dataset with background color highly correlated with each target class. The data of class $y$ has the background color of $b(y)$ with the probability of $\rho$ and has random background color among the rest colors with the probability of $(1 - \rho)$ [3]. Here, we use $\rho = 0.995$. We then observe the cosine distance between same/different bias class samples in the feature space of the bias-capturing model. As shown in Figure 2a, the same bias class sample pairs have near 0 distances while different bias class sample pairs have far distances. Based on this observation, we propose **Soft** Bias-**Con**trastive (SoftCon) loss for unknown bias label case that weights the BiasCon loss of each same class sample pair with a cosine distance in the feature space of the bias-capturing model as follows:

$$L_{\text{SoftCon}} = -\frac{1}{2N} \sum_{i \in I_{mv}} \frac{1}{C(i)} \sum_{j \in J(i)} d_{cos}(w(x_i), w(x_j)) \cdot \log \frac{\exp(z_i \cdot z_j / \tau)}{\sum_{a \in I_{mv} \setminus \{i\}} \exp(z_i \cdot z_a / \tau)}, \quad (11)$$

where $I_{mv}$, $z_i$ and $\tau$ inherit the definitions in BiasCon loss, $J(i) := \{j \in I_{mv} : y_j = y_i\}$, $w(x_i)$ is a feature embedding of sample $x_i$ extracted from the penultimate layer of the bias-capturing model, $d_{cos}(u, v) := 1 - \frac{u \cdot v}{\|u\| \|v\|}$ is a cosine distance and $C(i) = \sum_{j \in J(i)} d_{cos}(w(x_i), w(x_j))$. With the help of the bias-capturing model, the SoftCon loss in (11) can be utilized without the bias label information, but perhaps someone may be concerned that its performance is highly affected by the behaviors of the bias-capturing model. To address this concern, we empirically show in the Appendix D.1 that the SoftCon loss persists reasonable performance even when the bias-capturing model severely fails to capture the bias features only.

The SoftCon loss also suffers from the problem that a single batch may not contain enough positive pairs, i.e., pairs with far cosine distances. However, when the bias label is unavailable, we do not know $p(b|y)$. Instead, we define the bias-sparseness of the sample $x$ of class $y(x)$:

$$t(x) := \sum_{x' \in \mathcal{D}_{y(x)}} d_{cos}(w(x), w(x')), \quad (12)$$

where $\mathcal{D}_y$ is a set of class $y$ samples. If the sample $x$ is bias-sparse, only few same class samples have the same bias features, thus many positive pairs are available. Therefore, oversampling the training

Table 1: Unbiased accuracy and standard error evaluated on the BiasedMNIST [3] dataset with various target-bias correlations.

| Corr | Vanilla | LNL [28] | DI [43] | EnD [40] | **BiasCon** | **BiasBal** | **BC+BB** |
|------|---------|----------|---------|----------|-------------|-------------|-----------|
| 0.999 | $11.8_{\pm0.7}$ | $18.2_{\pm1.2}$ | $15.7_{\pm1.2}$ | $59.5_{\pm2.3}$ | $\mathbf{94.5_{\pm0.4}}$ | $76.8_{\pm1.6}$ | $94.0_{\pm0.6}$ |
| 0.997 | $62.5_{\pm2.9}$ | $57.2_{\pm2.2}$ | $60.5_{\pm2.2}$ | $82.7_{\pm0.3}$ | $97.0_{\pm0.0}$ | $91.2_{\pm0.2}$ | $\mathbf{97.3_{\pm0.1}}$ |
| 0.995 | $79.5_{\pm0.1}$ | $72.5_{\pm0.9}$ | $89.8_{\pm2.0}$ | $94.0_{\pm0.6}$ | $97.4_{\pm0.1}$ | $93.9_{\pm0.1}$ | $\mathbf{97.7_{\pm0.1}}$ |
| 0.99 | $90.8_{\pm0.3}$ | $86.0_{\pm0.2}$ | $96.9_{\pm0.1}$ | $94.8_{\pm0.3}$ | $97.7_{\pm0.1}$ | $96.3_{\pm0.2}$ | $\mathbf{98.1_{\pm0.1}}$ |
| 0.95 | $97.3_{\pm0.2}$ | $96.4_{\pm0.1}$ | $98.6_{\pm0.1}$ | $98.3_{\pm0.1}$ | $98.6_{\pm0.1}$ | $98.1_{\pm0.0}$ | $\mathbf{98.9_{\pm0.0}}$ |
| 0.9 | $98.2_{\pm0.1}$ | $94.0_{\pm0.3}$ | $98.8_{\pm0.1}$ | $98.7_{\pm0.0}$ | $98.9_{\pm0.1}$ | $98.5_{\pm0.1}$ | $\mathbf{99.3_{\pm0.0}}$ |

data based on the bias-sparseness gives the similar effect with oversampling based on $1/p(b|y)$ used in (4). In specific, we define the sampling frequency $Q(i)$ of the $i$-th sample as follows:

$$Q(i) \propto \begin{cases} t(x_i) & \text{if } t(x_i) < \gamma \cdot \min t(x) \\ \gamma \cdot \min t(x) & \text{o.w.} \end{cases}, \tag{13}$$

and $\gamma$ is the clipping hyperparameter. In practice, for each class $y$, we randomly select 1,024 samples and calculate $t(x)$ over those samples.

## 4 Experiments

We conduct experiments to evaluate how well our proposed method performs debiasing. To do this, we train the model with a biased dataset and measure the debiasing performance with various metrics. The main metric that we focus on is an unbiased accuracy [3, 33]. We first assess BiasCon and BiasBal loss on the dataset where the bias label is categorical and explicitly available. We conduct a controlled experiment on BiasedMNIST [3], where each digit is highly correlated with certain background color. We also evaluate on the real-world datasets CelebA [31] and UTKFace [46] where the dataset is biased toward sensitive attributes (e.g., gender, race). Next, we evaluate the SoftCon loss on BiasedMNIST but without bias label and 9-Class ImageNet [36, 24] that the CNN models tend to show the texture bias. We conduct all experiments with three different seeds and report the mean and standard error of each metric. Source code for our experiments is publicly available[1].

### 4.1 Performance when the bias label is available

We first evaluate our method using datasets with bias labels. In these datasets, each target class is highly correlated with a certain bias class and the vanilla model tends to use bias features in the prediction. We evaluate on BiasedMNIST [3], CelebA [31], and UTKFace [46]. Note that the bias label is unavailable at the test phase.

**Baselines** We compare our method with baselines that utilize the bias labels for training. [28] proposed a framework called a "Learning Not to Learn" (LNL), which uses an additional bias prediction branch to minimize the mutual information between the model feature and the bias label. [43] proposed a domain-independent (DI) training scheme that trains separated fully-connected head layers for each bias class and ensembles the outputs of each layer in the final prediction. EnD [40] uses a regularizer that disentangles the features of the same bias class samples.

**Controlled experiments** The first experiment to validate our method is a controlled experiment using BiasedMNIST [3] dataset. As explained in Section 3.4, BiasedMNIST is an MNIST [29] dataset that has colored background highly correlated with each target class. Vanilla CNN model conditions on the background color to predict the digit as it is easier to learn and achieve high accuracy. We use the correlation $\rho \in \{0.9, 0.95, 0.99, 0.995, 0.997, 0.999\}$ to evaluate the effectiveness and robustness of each method in various imbalance ratio. We explicitly construct a validation set and report the test unbiased accuracy at the epoch with the highest validation unbiased accuracy.

Table 1 shows the performance of each method in the BiasedMNIST dataset. Our BiasCon loss shows dramatic improvement in every $\rho$. Especially when the correlation gets very high to 0.999,

---

[1]https://github.com/grayhong/bias-contrastive-learning

Table 2: The unbiased / bias-conflict accuracy and standard error of the model trained on the CelebA [31] dataset.

| Target | Acc. Type | Vanilla | LNL [28] | DI [43] | EnD [40] | **BiasCon** | **BiasBal** | **BC+BB** |
|---|---|---|---|---|---|---|---|---|
| Blonde | Unbiased | $79.0_{\pm 0.1}$ | $80.1_{\pm 0.8}$ | $90.9_{\pm 0.3}$ | $86.9_{\pm 1.0}$ | $90.0_{\pm 0.2}$ | $91.1_{\pm 0.0}$ | $\mathbf{91.4_{\pm 0.0}}$ |
| | Bias-conflict | $59.0_{\pm 0.1}$ | $61.2_{\pm 1.5}$ | $86.3_{\pm 0.4}$ | $76.4_{\pm 1.9}$ | $85.1_{\pm 0.4}$ | $\mathbf{87.4_{\pm 0.2}}$ | $87.2_{\pm 0.2}$ |
| Makeup | Unbiased | $76.0_{\pm 0.8}$ | $76.4_{\pm 2.3}$ | $74.3_{\pm 1.1}$ | $74.8_{\pm 1.8}$ | $77.5_{\pm 0.7}$ | $76.0_{\pm 0.9}$ | $\mathbf{78.6_{\pm 1.8}}$ |
| | Bias-conflict | $55.2_{\pm 1.9}$ | $57.2_{\pm 4.6}$ | $53.8_{\pm 1.6}$ | $53.3_{\pm 3.6}$ | $61.3_{\pm 1.6}$ | $57.5_{\pm 2.1}$ | $\mathbf{63.5_{\pm 3.7}}$ |

Table 3: The unbiased / bias-conflict accuracy of the model trained on the UTKFace [46] dataset.

| Bias | Acc. Type | Vanilla | LNL [28] | DI [43] | EnD [40] | **BiasCon** | **BiasBal** | **BC+BB** |
|---|---|---|---|---|---|---|---|---|
| Race | Unbiased | $87.4_{\pm 0.3}$ | $87.3_{\pm 0.3}$ | $88.9_{\pm 1.2}$ | $88.4_{\pm 0.3}$ | $90.3_{\pm 0.2}$ | $90.4_{\pm 0.3}$ | $\mathbf{91.0_{\pm 0.2}}$ |
| | Bias-conflict | $79.1_{\pm 0.3}$ | $78.8_{\pm 0.6}$ | $89.1_{\pm 1.6}$ | $81.6_{\pm 0.3}$ | $88.8_{\pm 0.5}$ | $\mathbf{89.9_{\pm 0.6}}$ | $89.2_{\pm 0.1}$ |
| Age | Unbiased | $72.3_{\pm 0.3}$ | $72.9_{\pm 0.1}$ | $75.6_{\pm 0.8}$ | $73.2_{\pm 0.3}$ | $75.7_{\pm 0.2}$ | $78.8_{\pm 0.4}$ | $\mathbf{79.1_{\pm 0.3}}$ |
| | Bias-conflict | $46.5_{\pm 0.2}$ | $47.0_{\pm 0.1}$ | $60.0_{\pm 0.2}$ | $47.9_{\pm 0.6}$ | $61.7_{\pm 0.5}$ | $\mathbf{76.7_{\pm 3.2}}$ | $71.7_{\pm 0.8}$ |

our BiasCon loss still performs debiasing very well while the baselines fail to learn the target features, showing low unbiased accuracy. This shows the effectiveness and robustness of our BiasCon loss. Moreover, BiasBal loss itself shows higher performance than the previous state-of-the-art method. This indicates that simply optimizing toward uniform $p(y|b)$ indeed makes the bias features uninformative in solving the target task to some extent, thus prevents the model from learning those features. When two methods are jointly used (BC+BB in Table 1), the performance gets even better in most cases. This additional improvement implies that two methods are offering different aspects of debiasing.

**Real-world dataset** To evaluate our method in the real-world datasets, we consider CelebA [31] and UTKFace [46], as both datasets contain the meta-information about each data. For CelebA, we follow [33] and train the binary classification model where `HeavyMakeup` and `BlondHair` are target attributes and `Male` is a bias attribute. However, for `BlondHair` attribute, we find that both target classes are biased toward a non-`Male` bias class. Therefore, to ensure that the dataset is biased, for a `BlondHair` attribute task, we randomly select a subset of the dataset so that two classes are biased toward different bias classes. For UTKFace, we do the binary classification with `Gender` as a target attribute and `Race` or `Age` as bias attributes. We also truncate a portion of data to force the correlation between $Y$ and $B$ to be $p(y|b) = 0.9$. For both datasets, we use ImageNet-pretrained ResNet18 [22]. As previous experiments, we explicitly construct a validation set and report the test unbiased accuracy at the epoch with the highest validation unbiased accuracy. Details are available in the Appendix C.3. Here, in addition to the unbiased accuracy, we report the bias-conflict accuracy, which is an accuracy of minor subgroup samples.

Results in Table 2 show that our BiasCon and BiasBal loss both outperform the previous methods with large margin. We emphasize that both methods do not hurt the performance of bias-aligning samples as they also increase the total unbiased accuracy. Unlike the results in the controlled experiment, the BiasBal loss shows better performance than the BiasCon loss in the `BlondHair` attribute task. We do not claim the superiority of one method over the other between two methods. They give orthogonal effects in debiasing and the joint use of two methods (BC+BB) leads to further improvement. Similarly, our methods also show the superior performance in UTKFace, as shown in Table 3. As a result, the consistently superior results in two real-world datasets imply that both BiasCon and BiasBal losses successfully scale to real-world datasets.

## 4.2 Performance when the bias label is unavailable

We now extend to the cases where the bias labels are unavailable, and the type of the bias is beyond categorical data and evaluate our SoftCon loss. Even though the bias label is unavailable, we still assume that we have prior knowledge about the bias and we can design a bias-capturing model [11, 3].

Table 4: Unbiased accuracy and standard error evaluated on the BiasedMNIST [3] dataset without bias labels.

| Corr. | Vanilla | LM [11] | RUBi [8] | ReBias [3] | LfF [33] | SoftCon |
|---|---|---|---|---|---|---|
| 0.999 | $11.8_{\pm 1.1}$ | $10.5_{\pm 0.6}$ | $10.6_{\pm 0.5}$ | $26.5_{\pm 1.4}$ | $15.3_{\pm 2.9}$ | $\mathbf{65.0_{\pm 3.2}}$ |
| 0.997 | $57.2_{\pm 0.9}$ | $56.0_{\pm 4.3}$ | $49.6_{\pm 1.5}$ | $65.8_{\pm 0.3}$ | $63.7_{\pm 20.3}$ | $\mathbf{88.6_{\pm 1.0}}$ |
| 0.995 | $74.5_{\pm 1.4}$ | $80.9_{\pm 0.9}$ | $71.8_{\pm 0.5}$ | $75.4_{\pm 1.0}$ | $90.3_{\pm 1.4}$ | $\mathbf{93.1_{\pm 0.2}}$ |
| 0.99 | $88.9_{\pm 0.2}$ | $91.5_{\pm 0.4}$ | $85.9_{\pm 0.1}$ | $88.4_{\pm 0.6}$ | $95.1_{\pm 0.1}$ | $\mathbf{95.2_{\pm 0.4}}$ |
| 0.95 | $97.1_{\pm 0.0}$ | $93.6_{\pm 0.5}$ | $96.6_{\pm 0.1}$ | $97.0_{\pm 0.0}$ | $97.7_{\pm 0.2}$ | $\mathbf{98.0_{\pm 0.1}}$ |
| 0.9 | $98.2_{\pm 0.1}$ | $89.5_{\pm 0.7}$ | $97.8_{\pm 0.1}$ | $98.1_{\pm 0.1}$ | $96.1_{\pm 1.1}$ | $\mathbf{98.4_{\pm 0.1}}$ |

Table 5: Accuracy and standard error on 9-Class ImageNet [24] where bias labels are unavailable. Row with † denotes results directly borrowed from [3].

| Acc. Types | Vanilla | SIN† [18] | LM [11] | RUBi [8] | ReBias [3] | LfF [33] | SoftCon |
|---|---|---|---|---|---|---|---|
| Biased | $94.0_{\pm 0.1}$ | $88.4_{\pm 0.9}$ | $79.2_{\pm 1.1}$ | $93.9_{\pm 0.2}$ | $94.0_{\pm 0.2}$ | $91.2_{\pm 0.1}$ | $\mathbf{95.3_{\pm 0.2}}$ |
| Unbiased | $92.7_{\pm 0.2}$ | $86.6_{\pm 1.0}$ | $76.6_{\pm 1.2}$ | $92.5_{\pm 0.2}$ | $92.7_{\pm 0.2}$ | $89.6_{\pm 0.3}$ | $\mathbf{94.1_{\pm 0.3}}$ |
| IN-A | $30.5_{\pm 0.5}$ | $24.6_{\pm 2.4}$ | $19.0_{\pm 1.2}$ | $31.0_{\pm 0.2}$ | $30.5_{\pm 0.2}$ | $29.4_{\pm 0.8}$ | $\mathbf{34.1_{\pm 0.6}}$ |

**Baselines**   We compare debiasing methods that work without explicit bias labels. Learned-Mixin+H (LM) [11] and RUBi [8] utilize ensemble learning to promote the model to learn orthogonal features from the bias-capturing model, while ReBias [3] devises a regularizer that induces such learning. LfF [33] leverages the finding that bias-conflicting samples have high loss in the early stage of learning by weighting such samples. Additionally for the ImageNet, we compare with StylizedImageNet (SIN) [18] that randomizes the texture of ImageNet data to inhibit the model from conditioning on texture biases.

**Controlled experiments**   We use the same configuration with the controlled experiment of the known bias label case, explained in Section 4.1. We evaluate on BiasedMNIST [3], but now the bias label is unavailable. For all methods, we design the bias-capturing CNN model with a 1x1 filter. For the SoftCon loss, we pre-train the bias-capturing model for 80 epochs. Since we do not have access to the bias labels, we cannot construct a validation set. Thus, we report the final test unbiased accuracy.

In Table 4, we report the performance of each method on BiasedMNIST. Our SoftCon loss shows superior performance for all correlation cases. Even for the case with the highest correlation of 0.999, SoftCon loss fairly maintains its performance while previous methods severely fail. This indicates that our distance-based weighting successfully captures the positive pairs, i.e., sample pairs that need to be pulled closer. This also implies that even though the positive pair selection is not perfect, the concept of the bias-contrastive learning framework is robust enough to give a powerful debiasing effect.

**Real-world dataset**   We test our method on more realistic settings, 9-Class ImageNet [24], a subset of ImageNet [36] with 9 super-classes. Following [3], we aim to train the ResNet18 [22] model from scratch without texture bias which CNN models are susceptible to have. To achieve this, we use BagNet18 [6] as a bias-capturing model since it mainly uses a 1x1 kernel and has small receptive fields, and thus more susceptible to the local texture bias. To measure the unbiased accuracy, we use the texture bias label assigned by [3] and average the accuracy measured on each texture group. We also use ImageNet-A (IN-A) [23] as an additional test set as it contains bias-conflicting samples that vanilla CNN models fail to classify correctly. Due to the lack of bias labels in training set, we report results of the model after the full training of 120 epochs. Details are available in the Appendix C.5.

We report the performance of each method on 9-Class ImageNet in Table 5. Results show that our SoftCon loss shows the best unbiased and ImageNet-A accuracy. This indicates that the feature space of the bias-capturing model can give meaningful information about the bias even in the real-world scenario, and our SoftCon loss is effective and scalable.

# 5 Discussion and conclusion

In this work, we proposed the powerful debiasing method named Bias-Contrastive (BiasCon) loss, which effectively revises the contrastive learning framework for debiasing. We further improved the debiasing performance with Bias-Balanced (BiasBal) regression which minimizes the risk of the cross-entropy loss toward unbiased data distribution. To handle the case where the bias label is unavailable, we propose Soft Bias-Contrastive (SoftCon) loss by utilizing the feature space of the bias-capturing model. Experiments on various real-world datasets including CelebA, UTKFace and ImageNet show that our method significantly improves the previous debiasing approaches.

**Limitations of our work**    The first limitation of our work is that, as in [3], our work still assumes that we know the form of the bias (e.g., texture) and we can design the bias-capturing model that mostly uses the bias features in prediction. We believe that we can relax this assumption by considering the features learned earlier as bias features, as in [33]. Furthermore, it is challenging and interesting future work to adapt our model beyond the classification task such as VQA [1, 11].

**Societal impact**    As unbiasedness is the important property that machine learning models need to achieve, we expect our work gives a positive societal impact.

# Acknowledgement

This work was supported by the National Research Foundation of Korea (NRF) grants (2018R1A5A1059921, 2019R1C1C1009192) and Institute of Information & Communications Technology Planning & Evaluation (IITP) grants (No.2017-0-01779, A machine learning and statistical inference framework for explainable artificial intelligence, No.2019-0-01371, Development of brain-inspired AI with human-like intelligence, No.2019-0-00075, Artificial Intelligence Graduate School Program (KAIST)) funded by the Korea government (MSIT).

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
