# Appendix for "Unbiased Classification Through Bias-Contrastive and Bias-Balanced Learning"

**Youngkyu Hong**[†]
Naver AI Lab
youngkyu.hong@navercorp.com

**Eunho Yang**
KAIST
eunhoy@kaist.ac.kr

## A Proofs

### A.1 Proof of Theorem 1

We consider the data generation process that both target $Y$ and bias $B$ affect the generation of data $X$ as in Figure A1.

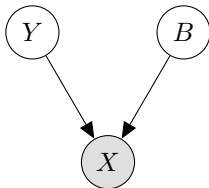

Figure A1: The data generation process of data $X$ where $B$ is a bias, and $Y$ is a target.

Our objective is to train the model to estimate the conditional probability over $P_u$ where $P_u$ is the data distribution shifted from $P_{train}$ to have the uniform correlation between Y and B (Section 3.3):

$$P_u(X, Y, B) \coloneqq P_{train}(X|Y, B)P_u(Y, B), \tag{A.1}$$

$$P_u(Y, B) \coloneqq P_u(Y|B)P_{train}(B) = \frac{1}{C} \cdot P_{train}(B). \tag{A.2}$$

**Theorem A.1.** *(Bias-balanced Regression) Assume the multinomial logistic regression with the model $h(x)[y] = \frac{e^{\eta_y}}{\sum_{y'} e^{\eta_{y'}}}$ where $\eta_y$ is the logit for class $y$, and the training data distribution $P_{train}(X, Y, B)$. Let $P_u$ be the data distribution shifted from $P_{train}$ to have the uniform correlation between Y and B. If the model $h$ estimates the conditional probability over $P_u$, i.e., $h(x)[y] = p_u(y|x)$, then the estimate of the conditional probability over $P_{train}$ is:*

$$p_{train}(y|x) = \frac{\exp\left(\eta_y + \log p_{train}(y|b)\right)}{\sum_{y'} \exp\left(\eta_{y'} + \log p_{train}(y'|b)\right)}, \tag{A.3}$$

*and the cross-entropy loss risk associated with $h$ over $P_{train}$ is $\mathbb{E}_{P_{train}}[L(h(x), y, b)]$, where $L(h(x), y, b)$ is defined as follows:*

$$L(h(x), y, b) \coloneqq -\log \frac{\exp\left(\eta_y + \log p_{train}(y|b)\right)}{\sum_{y'} \exp\left(\eta_{y'} + \log p_{train}(y'|b)\right)}. \tag{A.4}$$

*Proof.* In the data generation process in Figure A1, the conditional probability $p(y|x)$ can be decomposed using Bayesian interpretation:

$$p(y|x) = p(y|x, b) = \frac{p(x|y, b)}{p(x|b)} p(y|b), \tag{A.5}$$

---

[†]This work was done as a student at KAIST.

35th Conference on Neural Information Processing Systems (NeurIPS 2021).

where $b$ is a bias class of $x$. Here, $b$ is not an explicit input to the model but can be viewed as a latent variable in the conditional probability as $B$ entirely affects the generation of $X$.

For the notation simplicity, define $\phi_j$ and $\hat{\phi}_j$:

$$\phi_j := p_u(y{=}j|x,b) = \frac{e^{\eta_j}}{\sum_{i=1}^{C} e^{\eta_i}}, \ \hat{\phi}_j := p_{train}(y{=}j|x,b) = \frac{e^{\hat{\eta}_j}}{\sum_{i=1}^{C} e^{\hat{\eta}_i}}, \tag{A.6}$$

where $C$ is the number of target classes and $\hat{\eta}_j$ is the modified logit for the prediction on $P_{train}$ that we will later express using $\eta_j$. Then, using the canonical link function of the multinomial logistic regression,

$$\eta_j = \log \frac{\phi_j}{\phi_C}, \ \hat{\eta}_j = \log \frac{\hat{\phi}_j}{\hat{\phi}_C}. \tag{A.7}$$

Since we assume $p_u(x|y,b) = p_{train}(x|y,b)$, by using the decomposition in (A.5),

$$\hat{\phi}_j = \phi_j \cdot \frac{p_{train}(y{=}j|b)}{p_u(y{=}j|b)} \cdot \frac{p_u(x|b)}{p_{train}(x|b)}, \tag{A.8}$$

where $p_u(y{=}j|b) = \frac{1}{C}$. Thus, when we apply (A.8) to (A.7),

$$\hat{\eta}_j = \log \frac{\phi_j \cdot p_{train}(y{=}j|b) \cdot K}{\hat{\phi}_C}, \tag{A.9}$$

where $K$ is a constant that does not depend on $j$. Then, by combining (A.6) and (A.9),

$$\hat{\phi}_j = \frac{\phi_j \cdot p_{train}(y{=}j|b)}{\sum_{i=1}^{C} \phi_i \cdot p_{train}(y{=}i|b)} = \frac{\exp\left(\eta_j + \log p_{train}(y{=}j|b)\right)}{\sum_{i=1}^{C} \exp\left(\eta_i + \log p_{train}(y{=}i|b)\right)}. \tag{A.10}$$

Therefore, the cross-entropy loss risk associated with $h$ over $P_{train}$ is

$$\mathbb{E}_{P_{train}}[-\log p_{train}(y|x)] = \mathbb{E}_{P_{train}}[L(h(x),y,b)], \tag{A.11}$$

where $L(h(x),y,b)$ is defined as follows:

$$L(h(x),y,b) := -\log \frac{\exp\left(\eta_y + \log p_{train}(y|b)\right)}{\sum_{y'} \exp\left(\eta_{y'} + \log p_{train}(y'|b)\right)}. \tag{A.12}$$

$\square$

## A.2 Relationship between the BiasCon loss and EnD

Our BiasCon loss shares similar goal with EnD [17], but significantly improves the performance by replacing the shortcomings of the EnD. The regularization term used in EnD is as follows:

$$L = -\frac{1}{N} \sum_{i \in I} \left( \alpha \cdot \frac{1}{|P(i)|} \sum_{j \in P(i)} z_i \cdot z_j - \beta \cdot \frac{1}{|N(i)|} \sum_{j \in N(i)} z_i \cdot z_j \right), \tag{A.13}$$

where $\alpha$ and $\beta$ are weight hyperparameters, $I := \{1, ..., N\}$ is the index set of the sampled batch, $P(i) := \{j \in I : y_j = y_i, b_j \neq b_i\}$ is the index set of positive samples paired with $i$-th sample, $N(i) := \{j \in I : b_j = b_i\}$ is the index set of negative samples paired with $i$-th sample, $z_i = f(x_i)/\|f(x_i)\|$ is a normalized feature of $i$-th sample extracted from $f$. The most critical weakness of the EnD is that even though the model successfully learned the target features without bias features, the loss term still gives penalty to the negative pairs with the same biases (which are mostly the same target classes as bias and target are highly correlated).

For the BiasCon loss, if we omit the multiviewed batch, the loss is defined as

$$L = -\frac{1}{N} \sum_{i \in I} \frac{1}{|J(i)|} \sum_{j \in J(i)} \log \frac{\exp\left(z_i \cdot z_j / \tau\right)}{\sum_{a \in I \setminus \{i\}} \exp\left(z_i \cdot z_a / \tau\right)}, \tag{A.14}$$

where $J(i) := \{j \in I : y_j = y_i, b_j \neq b_i\}$ is the index set of positive samples paired with $i$-th sample and $\tau$ is a temperature hyperparameter. Then, as shown in [18], we can give the following

approximation about the contrastive loss on $j$-th positive sample paired with $i$-th sample by applying the limit $\tau \to 0^+$.

$$\lim_{\tau \to 0^+} -\log \frac{\exp\left(z_i \cdot z_j / \tau\right)}{\sum_{a \in I \setminus \{i\}} \exp\left(z_i \cdot z_a / \tau\right)} \tag{A.15}$$

$$= \lim_{\tau \to 0^+} \log \left[ 1 + \sum_{a \neq j} \exp\left((z_i \cdot z_a - z_i \cdot z_j)/\tau\right) \right] \tag{A.16}$$

$$= \lim_{\tau \to 0^+} \log \left[ 1 + \sum_{z_i \cdot z_a \geq z_i \cdot z_j} \exp\left((z_i \cdot z_a - z_i \cdot z_j)/\tau\right) \right] \tag{A.17}$$

$$= \lim_{\tau \to 0^+} \frac{1}{\tau} \max_a [z_i \cdot z_a - z_i \cdot z_j, 0]. \tag{A.18}$$

Therefore, the BiasCon loss gives penalty if there exist negative pairs that are closer than positive pairs. In other words, we can roughly argue that if the model successfully learned target features without bias features so that there is no preference among the same target classes with or without the same bias features in terms of cosine distance, then the BiasCon loss is 0. This way, the shortcomings of the EnD can be solved. The results in Table A11 show that the BiasCon loss without multiviewed batch or oversampling strategy (row named BiasCon − Aug. − Samp.) outperforms EnD in BiasedMNIST [1] dataset in high correlation settings. In conclusion, the BiasCon loss improves over the EnD by fixing the shortcomings explained above.

### A.3   Intuition behind BiasCon and SoftCon loss

Using the fact that the contrastive loss follows the formulation of InfoNCE [14], the estimator of the mutual information, we can give a probabilistic interpretation of the BiasCon loss. For the brevity, we omit the multiviewed batch used in the BiasCon loss. Then, the formulation of the BiasCon loss is as follows:

$$L = -\frac{1}{N} \sum_{i \in I} \frac{1}{|J(i)|} \sum_{j \in J(i)} \log \frac{\exp\left(z_i \cdot z_j / \tau\right)}{\sum_{a \in I \setminus \{i\}} \exp\left(z_i \cdot z_a / \tau\right)}, \tag{A.19}$$

where $I := \{1, ..., N\}$ is the index set of the sampled batch, $J(i) := \{j \in I : y_j = y_i, b_j \neq b_i\}$ is the index set of positive samples paired with $i$-th sample, $z_i = f(x_i)/\|f(x_i)\|$ is a normalized feature of $i$-th sample extracted from $f$, the network up to the penultimate layer, and $\tau$ is a temperature hyperparameter. Now, we consider the function $u(z_1, z_2)$ where $u = 1$ if $x_1$ and $x_2$ have same target class but different bias classes and the probability function $P(z_1, z_2) = \frac{u(z_1, z_2)}{\sum_{(i,j)} u(z_{1i}, z_{2j})}$. Then, if we define the random variable $(Z_1, Z_2) \sim P$, InfoNCE estimator of $I(Z_1; Z_2)$, is defined as

$$I(Z_1; Z_2) \geq \mathbb{E}\left[ \frac{1}{N} \sum_{i=1}^{N} \log \frac{h(z_{1i}, z_{2i})}{\frac{1}{N} \sum_j h(z_{1i}, z_{2j})} \right], \tag{A.20}$$

where the expectation is over $N$ independent samples $(z_{1i}, z_{2j})$ from $(Z_1, Z_2)$, and $h$ is the critic function. If we consider the critic function $h(z_1, z_2) := \exp\left(z_i \cdot z_j / \tau\right)$, we can see that the absolute value of the BiasCon loss is the same as InfoNCE estimator of $I(Z_1; Z_2)$ with a constant difference. Thus, we can interpret that minimizing the BiasCon loss is equal to inducing the model to maximize the mutual information $I(Z_1; Z_2)$, which are the normalized features of samples from the same target class but different bias classes. Therefore, the feature extractor $f$ learns to extract the target features while discarding the bias features.

We can give a similar probabilistic interpretation of the SoftCon loss. Likewise, we omit the multiviewed batch used in the SoftCon loss. Then, the formulation of the SoftCon loss is

$$L = -\frac{1}{N} \sum_{i \in I} \frac{1}{C(i)} \sum_{j \in J(i)} d_{cos}(w(x_i), w(x_j)) \cdot \log \frac{\exp\left(z_i \cdot z_j / \tau\right)}{\sum_{a \in I} \exp\left(z_i \cdot z_a / \tau\right)}, \tag{A.21}$$

where $I$, $z_i$ and $\tau$ inherit the definitions in BiasCon loss, $J(i) := \{j \in I : y_j = y_i\}$, $w(x_i)$ is a feature embedding of sample $x_i$ extracted from the penultimate layer of the bias-capturing model,

$d_{cos}(u, v) := 1 - \frac{u \cdot v}{\|u\|\|v\|}$ is a cosine distance and $C(i) = \sum_{j \in J(i)} d_{cos}(w(x_i), w(x_j))$. Similar to the BiasCon loss case, we consider the function $u(z_1, z_2) \propto d_{cos}(w(x_1), w(x_2))$ only for $x_1$ and $x_2$ of the same target class and the probability function $P(z_1, z_2) = \frac{u(z_1, z_2)}{\sum_{(i,j)} u(z_{1i}, z_{2j})}$. Then, if we define the random variable $(Z_1, Z_2) \sim P$, InfoNCE estimator of $I(Z_1; Z_2)$, is defined as

$$I(Z_1; Z_2) \geq \mathbb{E}_P \left[ \frac{1}{N} \sum_{i=1}^{N} \log \frac{h(z_{1i}, z_{2i})}{\frac{1}{N} \sum_j h(z_{1i}, z_{2j})} \right] \tag{A.22}$$

$$\propto \mathbb{E}_{\text{Unif}} \left[ \frac{1}{N} \sum_{i=1}^{N} d_{cos}(w(x_{1i}), w(x_{2i})) \cdot \log \frac{h(z_{1i}, z_{2i})}{\frac{1}{N} \sum_j h(z_{1i}, z_{2j})} \right], \tag{A.23}$$

where the first expectation is over $N$ independent samples $(z_{1i}, z_{2j})$ from $(Z_1, Z_2)$, the second expectation is over $N$ independent samples $(z(x_{1i}, z(x_{2j}))$ from $(X_1, X_2) \sim$ Unif, Unif is a uniform distribution over training samples, and $h$ is the critic function. If we again consider the critic function $h(z_1, z_2) := \exp(z_i \cdot z_j / \tau)$, we can see that the absolute value of the SoftCon loss is the same as InfoNCE estimator of $I(Z_1; Z_2)$ with a constant difference. Thus, we can similarly interpret that minimizing the SoftCon loss is equal to inducing the model to maximize the mutual information $I(Z_1; Z_2)$. Therefore, the feature extractor $f$ learns to extract the features that are orthogonal to the bias-capturing model, i.e., if two samples are close in the feature space of the bias-capturing model, then two samples are far from each other in the feature space of the main model.

## B  Full algorithm

We show the full algorithm of training with the BiasCon and BiasBal loss for the known bias label case and training with SoftCon loss for the unknown case.

---

**Algorithm 1** Training with Bias-Contrastive loss

---

**Input:** Dataset $\mathcal{D} = \{(x_i, y_i, b_i)\}$, number of classes $N_c$, number of bias classes $N_b$, random augmentation $a(x)$, batch size $N$ and training epochs $T$.
**Output:** Debiased model $h_{\theta^T}$
For each $y \in \{1, \ldots, N_c\}$ and $b \in \{1, \ldots, N_b\}$, calculate $p(b|y)$
Calculate sampling frequency $Q(i)$ for each $i$-th sample
Initialize $\theta^0$
**for** $t \leftarrow 1$ **to** $T$ **do**
    $\mathcal{D}_{\text{CE}}^t \leftarrow \{x_i : x_i \sim \text{Unif}(\mathcal{D})\}$
    $\mathcal{D}_{\text{BC}}^t \leftarrow \{x_i : x_i \sim Q(i)\}$
    $\mathcal{D}_{\text{MV}}^t \leftarrow \bigcup_{j=1}^{2} \{a(x_i) : x_i \in \mathcal{D}_{\text{BC}}^t\}$
    $L \leftarrow L_{\text{CE}}(\mathcal{D}_{\text{CE}}^t) + L_{\text{BiasCon}}(\mathcal{D}_{\text{MV}}^t)$
    $\theta^t \leftarrow \theta^{t-1} - \eta \nabla_\theta L$
**end for**

---

---

**Algorithm 2** Bias-Balanced Regression

---

**Input:** Dataset $\mathcal{D} = \{(x_i, y_i, b_i)\}$, number of classes $N_c$, number of bias classes $N_b$, batch size $N$ and training epochs $T$.
**Output:** Debiased model $h_{\theta^T}$
For each $y \in \{1, \ldots, N_c\}$ and $b \in \{1, \ldots, N_b\}$, calculate $p(y|b)$
Initialize $\theta^0$
**for** $t \leftarrow 1$ **to** $T$ **do**
    $\mathcal{D}^t \leftarrow \{x_i : x_i \sim \text{Unif}(\mathcal{D})\}$
    $\theta^t \leftarrow \theta^{t-1} - \eta \nabla_\theta L_{\text{BiasBal}}(\mathcal{D}^t)$
**end for**

---

**Algorithm 3** Training with Soft Bias-Contrastive loss

---

**Input:** Dataset $\mathcal{D} = \{(x_i, y_i)\}$, bias-capturing model $g_{\theta_B}$, bias-capturing model training epochs $T_B$, anchor samples $N_a$, number of classes $N_c$, random augmentation $a(x)$, batch size $N$ and training epochs $T$.
**Output:** Debiased model $h_{\theta^T}$
Initialize $\theta_B^0$
**for** $t \leftarrow 1$ **to** $T_B$ **do**
 $\mathcal{D}^t \leftarrow \{x_i : x_i \sim \text{Unif}(\mathcal{D})\}$
 $\theta_B^t \leftarrow \theta_B^{t-1} - \eta \nabla_{\theta_B} L_{CE}(\mathcal{D}^t)$
**end for**
**for** $y \in \{1, \ldots, N_c\}$ **do**
 Randomly select $N_a$ anchor samples $\{x_1, \ldots, x_{N_a}\}$ from $\mathcal{D}_y$ for calculating $t(x)$
 **for** $x \in \mathcal{D}_y$ **do**
  $t(x) \leftarrow \sum\limits_{i=1}^{N_a} d_{cos}(w(x), w(x_i))$
 **end for**
**end for**
Calculate sampling frequency $Q(i)$ for each $i$-th sample using $t(x)$
Initialize $\theta^0$
**for** $t \leftarrow 1$ **to** $T$ **do**
 $\mathcal{D}_{CE}^t \leftarrow \{x_i : x_i \sim \text{Unif}(\mathcal{D})\}$
 $\mathcal{D}_{BC}^t \leftarrow \{x_i : x_i \sim Q(i)\}$
 $\mathcal{D}_{MV}^t \leftarrow \bigcup\limits_{j=1}^{2} \{a(x_i) : x_i \in \mathcal{D}_{BC}^t\}$
 $L \leftarrow L_{CE}(\mathcal{D}_{CE}^t) + L_{SoftCon}(\mathcal{D}_{MV}^t)$
 $\theta^t \leftarrow \theta^{t-1} - \eta \nabla_\theta L$
**end for**

---

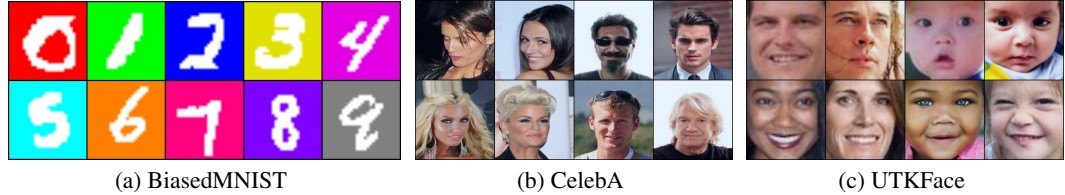

(a) BiasedMNIST     (b) CelebA     (c) UTKFace

Figure A2: Example samples of BiasedMNIST [1], CelebA [10] and UTKFace [22]. (a) In BiasedM-NIST, each digit is correlated with certain background color. (b) CelebA with target `Blonde` and bias `Gender`. (c) UTKFace with target `Gender` and bias `Age`.

## C   Experimental configurations

In this section, we explain the details of the experimental configurations used for all experiments in Section 4. Experiments are conducted with three different seeds and with the NVIDIA V100 GPU. We use Naver Smart Machine Learning (NSML) platform [16] to manage experiments.

### C.1   BiasedMNIST for known bias label case

We use the BiasedMNIST implementation of [1]. The data of class $y$ has the background color of $b(y)$ with the probability of $\rho$ and has random background color among the rest colors with the probability of $(1-\rho)$. The example sample of each class with its biased background color is shown in Figure A2a. The size of the image is $28 \times 28$, and the batch size is 128. For the BiasCon loss, we first sample the batch of size 64 and generate multiviewed batch with random augmentation of (a) random rotation with maximum $20°$, (b) random color jitter (change in brightness, contrast and saturation) with the factor of $0.4$ and with the probability of $0.8$. Note that we only apply the augmentation for the generation of multiviewed batch (Algorithm 1). The base architecture we use for the experiment

Table A1: Network architecture used for BiasedMNIST [1]. The kernel is written in the order of $H \times W \times C$. We use $k = 7$, $p = 2$ for the main model and $k = 1$, $p = 0$ for the bias-capturing model where $k$ is the kernel size and $p$ is the padding size used for the convolution.

| Layer | Kernel | Padding | Batch Norm | Activation |
|-------|--------|---------|------------|------------|
| Conv | $k \times k \times 16$ | $p$ | ◯ | ReLU |
| Conv | $k \times k \times 32$ | $p$ | ◯ | ReLU |
| Conv | $k \times k \times 64$ | $p$ | ◯ | ReLU |
| Conv | $k \times k \times 128$ | $p$ | ◯ | ReLU |
| AvgPool | - | - | - | - |
| FC | - | - | - | Softmax |

Table A2: Number of training samples used for CelebA [10].

| | | BlondHair | | HeavyMakeup | |
|------|---|------|-------|-------|-------|
| | | 0 | 1 | 0 | 1 |
| Male | 0 | 1558 | 18279 | 25789 | 49804 |
| | 1 | 53577 | 1098 | 54460 | 163 |

Table A3: Number of training samples used for UTKFace [22].

| | | Age | | Race | |
|--------|---|------|------|------|------|
| | | 0 | 1 | 0 | 1 |
| Gender | 0 | 8229 | 822 | 4354 | 435 |
| | 1 | 134 | 1346 | 534 | 5344 |

is explained in Table A1, where we use the kernel size of $k = 7$ and the padding size of $p = 2$. We use the Adam [9] optimizer with the learning rate of 0.001 for BiasCon loss and 0.0001 for BiasBal loss, hyperparameters $\beta_1 = 0.9, \beta_2 = 0.999$ and weight decay rate of 0.0001. We decay the learning rate by 0.1 at 1/3 and 2/3 of the total training epochs. All experiments are trained for 80 epochs. We use the temperature $\tau = 0.07$ for all experiments and the rest of the hyperparameters are reported in Table A4 for the BiasCon loss and in Table A5 for the joint use of BiasCon and BiasBal losses. We randomly sample 10000 images from training set and use random background color (without correlation) to construct validation set. The average training time with NVIDIA V100 GPU is 36m 1s for the BiasBal loss and 58m 59s for the BiasCon loss.

## C.2   CelebA

Following the experimental configuration of [13], we train a binary classifier that classifies whether the input image has the target attribute or not. We use the HeavyMakeup and BlondHair attributes as target attributes and Male as a bias attribute. However, for the task of classifying BlondHair attribute, both bias classes are skewed toward the BlondHair attribute. This means that the model cannot decide whether the class has BlondHair attribute or not by looking at the bias features. Thus, we intentionally truncate a part of the dataset and make each target class to be biased toward a certain bias class. The number of training samples used for each task is available in Table A2. The size of the image is $224 \times 224$, and the batch size is 128. For the BiasCon loss, we first sample the batch with the size 64 and apply additional random augmentations to generate the multiviewed batch (Algorithm 1). By default, we apply the augmentation of random flip. For the BiasCon loss, we additionally apply (a) random resized crop, (b) random color jitter (change in brightness, contrast, and saturation) with the factor of $0.4$ and with the probability of $0.8$, and (c) random grayscale conversion with the probability of $0.2$. Note that we only apply the additional augmentation for the generation of the multiviewed batch. We use the ResNet18 [4] model pretrained on the ImageNet [15] dataset. We use the Adam [9] optimizer with the learning rate of 0.001, hyperparameters of $\beta_1 = 0.9, \beta_2 = 0.999$, and weight decay rate of 0.0001. For the HeavyMakeup attribute classification task, we train the model for 40 epochs and BlondHair attribute classification task, we train the model for 10 epochs as the model converges much faster in this task. We decay the learning rate by 0.1 at 1/3 and 2/3 of the total training epochs. We use the temperature $\tau = 0.07$ for all experiments. We report the hyperparameters in Table A4 for the BiasCon loss, and in Table A5 for the joint use of BiasCon and BiasBal losses. The average training time with NVIDIA V100 GPU is 2h 23m 37s / 21m 7s for the BiasBal loss and 4h 16m 19s / 37m 32s for the BiasCon loss for HeavyMakeup / Blonde attribute classification task, respectively.

Table A4: Hyperparameter $\alpha$ and $\gamma$ used for the BiasCon loss.

| Param | BiasedMNIST | | | | | | CelebA | | UTKFace | |
|---|---|---|---|---|---|---|---|---|---|---|
| | 0.999 | 0.997 | 0.995 | 0.99 | 0.95 | 0.9 | Blonde | Makeup | Race | Age |
| $\alpha$ | 0.01 | 0.01 | 0.01 | 0.01 | 0.01 | 0.01 | 0.01 | 0.01 | 0.01 | 0.01 |
| $\gamma$ | 10 | 10 | 10 | 10 | 10 | 10 | 30 | 50 | 10 | 10 |

Table A5: Hyperparameter $\alpha$ and $\gamma$ used for the joint use of BiasCon and BiasBal losses.

| Param | BiasedMNIST | | | | | | CelebA | | UTKFace | |
|---|---|---|---|---|---|---|---|---|---|---|
| | 0.999 | 0.997 | 0.995 | 0.99 | 0.95 | 0.9 | Blonde | Makeup | Race | Age |
| $\alpha$ | 1 | 1 | 1 | 1 | 1 | 1 | 1 | 1 | 1 | 1 |
| $\gamma$ | 10 | 10 | 10 | 10 | 10 | 10 | 30 | 50 | 10 | 10 |

Table A6: Hyperparameter $\alpha$ and $\gamma$ used for the SoftCon loss.

| Param | BiasedMNIST | | | | | | ImageNet |
|---|---|---|---|---|---|---|---|
| | 0.999 | 0.997 | 0.995 | 0.99 | 0.95 | 0.9 | - |
| $\alpha$ | 0.01 | 0.01 | 0.01 | 0.01 | 0.01 | 0.01 | 1 |
| $\gamma$ | 50 | 50 | 50 | 50 | 50 | 50 | 0 |

### C.3 UTKFace

We also experiment on the UTKFace [22] where it has meta-information about the Race, Age, and Gender of the person in the image. We first assign bias classes using Age and Race attributes. Specifically, for the Age attribute, we divide samples into two groups; bias class 0 for samples with age $\geq 20$ and bias class 1 for samples with age $\leq 10$. For the Race attribute, we also divide samples into two groups; bias class 0 for samples with race=white and bias class 1 for samples with race $\neq$ white. The target class is divided using Gender attribute, where target class 0 is male, and target class 1 is female. We then construct a biased dataset for the gender classification task, which is biased toward Age or Race bias class by truncating a portion of samples. The number of training samples used for each task is available in Table A3. The size of the image is $64 \times 64$, and the batch size is 128. By default, we apply the augmentation of (a) random resized crop and (b) random flip. For the BiasCon loss, we first sample the batch with the size of 64 and generate the multiviewed batch by additionally applying (a) random color jitter (change in brightness, contrast, and saturation) with the factor of $0.4$ and with the probability of $0.8$, and (b) random grayscale conversion with the probability of $0.2$ (Algorithm 1). We use the ResNet18 [4] model pretrained on the ImageNet [15] dataset. Note that we only apply the additional augmentation for the generation of multiviewed batch (Algorithm 1). We use the Adam [9] optimizer with the learning rate of 0.001, hyperparameters of $\beta_1 = 0.9, \beta_2 = 0.999$, and weight decay rate of $0.0001$. For both tasks, we train the model for 20 epochs. We decay the learning rate by 0.1 at 1/3 and 2/3 of the total training epochs. We use the temperature $\tau = 0.07$. We report the hyperparameters in Table A4 for the BiasCon loss, and in Table A5 for the joint use of BiasCon and BiasBal losses. The average training time with NVIDIA V100 GPU is 3m 24s / 4m 4s for the BiasBal loss and 6m 3s / 5m 31s for the BiasCon loss for Age / Race bias attribute task, respectively.

### C.4 BiasedMNIST for unknown bias label case

We mostly follow the experimental setup explained in C.1. However, instead of using the bias label, we utilize the auxiliary bias-capturing model with the same architecture in Table A1 but with the kernel size $k = 1$ and the padding size $p = 0$. In the training with SoftCon loss, the bias-capturing model is pretrained for 80 epochs. In addition, for the sampling frequency calculation, we randomly

select 1,024 samples for each class as explained in Section 3.4 (Algorithm 3). We use the temperature $\tau = 0.07$ and the rest of the hyperparameters are shown in Table A6. The average training time with NVIDIA V100 GPU is 1h 1s for the SoftCon loss.

### C.5 ImageNet

We use the 9-Class ImageNet [6] with 9 super-classes. Most of the experimental configurations follow [1]. We use the ResNet18 [4] model without any pretraining. For the auxiliary bias-capturing model, we use the BagNet18 [2]. In the training with SoftCon loss, we pretrain the bias-capturing model for 120 epochs (Algorithm 3). Following [1], we use the Adam [9] optimizer with learning rate of 0.001, hyperparameters $\beta_1 = 0.9, \beta_2 = 0.999$, and weight decay rate of 0.0001 with cosine annealing learning rate scheduler [11]. The size of the image is $224 \times 224$, and the batch size is 128. By default, we apply the augmentation of (a) random resized crop, (b) random flip. For the SoftCon loss, we additionally apply (a) random color jitter (change in brightness, contrast, and saturation) with the factor of 0.4 and with the probability of 0.8, and (b) random grayscale conversion with the probability of 0.2. Note that we only apply the additional augmentation for the generation of the multiviewed batch (Algorithm 3). For all methods, we train the model for 120 epochs. We use the temperature $\tau = 0.07$ and Table A6 shows the rest of the hyperparameters used for the SoftCon loss. Interestingly, the oversampling with sampling frequency $Q(i)$ does not help for the ImageNet training, thus we do not apply the oversampling in this experiment. For the calculation of the unbiased accuracy, we use the texture bias labels directly copied from [1][1]. We also use the ImageNet-A [5] dataset as an additional test set. The average training time with NVIDIA V100 GPU is 7h 23m 34s for the SoftCon loss.

## D   Additional experiments

### D.1   Robustness of the SoftCon loss on the performance of the bias-capturing model

We evaluate the dependency of the SoftCon loss to the performance of the bias-capturing model with the BiasedMNIST [1] dataset. To check this, we first vary the performance of the bias-capturing model by constructing deteriorated bias-capturing model with the same architecture to Table A1, but with the kernel size of $k = 7$ and the padding size of $p = 2$. Large kernel size prevents the model from focusing only on the local feature, deteriorating the bias-capturing ability. The performance of the deteriorated bias-capturing model is measured by the bias accuracy. Bias accuracy is an accuracy measured with label $y^{-1}(b)$, the target class that is highly correlated with the bias class of the sample. If the model only uses the bias features in the prediction, the model will output the highly correlated target class $y^{-1}(b)$ for the input of samples with the bias class $b$, and thus the bias accuracy will be high. To vary the bias-capturing performance of the deteriorated bias-capturing model, we train the model with different target-bias correlation $\rho \in \{0.99, 0.995, 0.997\}$ of BiasedMNIST dataset. The bias-capturing performance of the model will be better when it is trained with a higher target-bias correlation. Here, we report unbiased accuracy of the final model after the full training.

Table A7 shows the performance of the SoftCon loss trained based on the deteriorated bias-capturing models. As shown in the second row of Table A7, the bias accuracy of the deteriorated bias-capturing model is 53.0% when it is trained on the dataset with $\rho = 0.997$ and gets down to 21.4% when trained on the dataset with $\rho = 0.99$. For the highly correlated cases with $\rho \in \{0.997, 0.999\}$, even if the bias-capturing model is worst broken, the SoftCon loss still retains an unbiased accuracy of 43.2%, much higher than the previous baselines. This indicates that our SoftCon loss is robust to the noise originated from the failure of the bias-capturing model. However, when the dataset bias is less severe, the model suffers from the poor bias-capturing ability. We expect that this is because the same high weight is applied to the SoftCon loss even when the bias-capturing model is corrupted. We believe that performing a hyperparameter search by constructing a small bias-labeled validation set can address this issue.

---

[1]https://github.com/clovaai/rebias

Table A7: Performance dependency of the SoftCon loss on the performance of the bias-capturing model. We measure the unbiased accuracy evaluated on the BiasedMNIST [1] dataset with various target-bias correlations. Performance of the deteriorated bias-capturing model is measured by the bias accuracy and reported in the second row (values under 'Bias-capturing model accuracy'). Reported values are mean and standard error over three independent runs with different seeds.

| Corr | ReBias [1] | LfF [13] | SoftCon | Bias-capturing model accuracy | | |
| --- | --- | --- | --- | --- | --- | --- |
| | | | | 53.0 | 34.7 | 21.4 |
| 0.999 | $26.5_{\pm1.4}$ | $15.3_{\pm2.9}$ | $\mathbf{65.0_{\pm3.2}}$ | $47.8_{\pm1.2}$ | $41.3_{\pm0.6}$ | $43.2_{\pm1.0}$ |
| 0.997 | $65.8_{\pm0.3}$ | $63.7_{\pm20.3}$ | $\mathbf{88.6_{\pm1.0}}$ | $68.8_{\pm1.9}$ | $64.3_{\pm0.7}$ | $58.5_{\pm1.1}$ |
| 0.995 | $75.4_{\pm1.0}$ | $90.3_{\pm1.4}$ | $\mathbf{93.1_{\pm0.2}}$ | $79.1_{\pm2.4}$ | $73.0_{\pm1.3}$ | $68.9_{\pm1.8}$ |
| 0.99 | $88.4_{\pm0.6}$ | $95.1_{\pm0.1}$ | $\mathbf{95.2_{\pm0.4}}$ | $91.0_{\pm1.3}$ | $85.6_{\pm2.1}$ | $85.2_{\pm0.6}$ |
| 0.95 | $97.0_{\pm0.0}$ | $97.7_{\pm0.2}$ | $\mathbf{98.0_{\pm0.1}}$ | $96.8_{\pm0.3}$ | $94.3_{\pm0.9}$ | $92.0_{\pm1.2}$ |
| 0.9 | $98.1_{\pm0.1}$ | $96.1_{\pm1.1}$ | $\mathbf{98.4_{\pm0.1}}$ | $98.0_{\pm0.2}$ | $96.0_{\pm1.9}$ | $96.6_{\pm0.4}$ |

Table A8: Comparison of the design choice of separately and jointly using the BiasCon loss with the cross-entropy loss. Design type 'Separate' means the separate training of the feature representation and the classifier head as in [8], and 'Joint' means our proposed way of using the BiasCon loss. We report the unbiased accuracy and standard error of two methods on the BiasedMNIST [1] dataset.

| Design type | 0.999 | 0.997 | 0.995 | 0.99 | 0.95 | 0.9 |
| --- | --- | --- | --- | --- | --- | --- |
| Separate | $85.4_{\pm1.0}$ | $90.1_{\pm0.8}$ | $92.4_{\pm0.9}$ | $93.4_{\pm1.2}$ | $97.9_{\pm0.1}$ | $93.7_{\pm0.5}$ |
| **Joint (Ours)** | $\mathbf{94.5_{\pm0.4}}$ | $\mathbf{97.0_{\pm0.0}}$ | $\mathbf{97.4_{\pm0.1}}$ | $\mathbf{97.7_{\pm0.1}}$ | $\mathbf{98.6_{\pm0.1}}$ | $\mathbf{98.9_{\pm0.1}}$ |

### D.2 Comparison of performance according to design choice of BiasCon loss

In [8], the SupCon loss is first solely used in feature representation training, then the fully-connected (FC) layer head is trained with the cross-entropy loss upon the frozen feature representation. However, we use the BiasCon loss as a regularizer that is jointly used with the cross-entropy loss so that the signal to incorporate all of the same target class samples is given during the feature representation training. Here, we compare the performance of two design choices on BiasedMNIST [1].

We report the performance of two design choices in Table A8. The separate use of the BiasCon and cross-entropy losses shows much worse unbiased accuracy in all correlations. This shows that the BiasCon loss helps the debiasing, but itself is not powerful enough to train a good feature representation. The result validates the design choice of ours, which uses the BiasCon loss jointly with the cross-entropy loss.

We further investigate on the loss design of the BiasCon loss. As explained in Section A.2, the BiasCon loss regularizes the model from having a feature space that same target and same bias samples are closer than the same target and different bias samples. To achieve this, we set the same target and same bias samples as a negative pairs of the contrastive loss. However, some might concern that this might exacerbate the effect of the cross-entropy loss on learning target features. To address this concern, we assess some variants that can induce similar effect.

- Loss variant 1: positive pair - same class / negative pair - different class but same bias
- Loss variant 2: positive pair - same class but different bias / negative pair - different class
- Loss varaint 3: Train a feature representation with two contrastive losses; Loss 1: positive pair - same class / negative pair - diff class, Loss 2: positive pair - different bias / negative pair - same bias, then train the classifier head upon the frozen representation.

Table A9 shows the results of each loss variant on BiasedMNIST dataset. Poor performance of the first loss variant is expected since it cannot give enough penalty as positive pairs contain most of the same bias samples (as targets and biases are highly correlated) while the negative pairs contain much fewer same bias samples. The second loss variant removes the same class samples from the negative pairs. This loss shows a debiasing effect to some extent as it induces the same class but different bias

Table A9: Comparison of the alternative loss design choices of the BiasCon loss. We report the unbiased accuracy and standard error of two methods on the BiasedMNIST [1] dataset with various target-bias correlations.

| Loss type | 0.999 | 0.997 | 0.995 | 0.99 |
|---|---|---|---|---|
| Variant 1 | $11.6_{\pm0.2}$ | $20.5_{\pm1.5}$ | $26.8_{\pm2.9}$ | $35.7_{\pm3.0}$ |
| Variant 2 | $81.2_{\pm1.5}$ | $83.8_{\pm0.1}$ | $85.3_{\pm1.3}$ | $85.9_{\pm2.2}$ |
| Variant 3 | $60.3_{\pm1.2}$ | $77.9_{\pm0.4}$ | $86.1_{\pm0.8}$ | $89.9_{\pm0.8}$ |
| **BiasCon** | $\mathbf{94.5_{\pm0.4}}$ | $\mathbf{97.0_{\pm0.0}}$ | $\mathbf{97.4_{\pm0.1}}$ | $\mathbf{97.7_{\pm0.1}}$ |

Table A10: Unbiased accuracy and standard error of baseline trained with/without BiasBal loss on BiasedMNIST [1].

| Method | BiasedMNIST | | | | | |
|---|---|---|---|---|---|---|
| | 0.999 | 0.997 | 0.995 | 0.99 | 0.95 | 0.9 |
| ReBias [1] | $26.5_{\pm1.4}$ | $65.8_{\pm0.3}$ | $75.4_{\pm1.0}$ | $88.4_{\pm0.6}$ | $97.0_{\pm0.0}$ | $98.1_{\pm0.1}$ |
| **ReBias + BB** | $\mathbf{45.5_{\pm2.3}}$ | $\mathbf{88.1_{\pm0.9}}$ | $\mathbf{92.5_{\pm0.2}}$ | $\mathbf{95.5_{\pm0.2}}$ | $\mathbf{98.0_{\pm0.1}}$ | $\mathbf{98.5_{\pm0.1}}$ |
| EnD [17] | $\mathbf{59.5_{\pm2.3}}$ | $82.7_{\pm0.3}$ | $\mathbf{94.0_{\pm0.6}}$ | $\mathbf{94.8_{\pm0.3}}$ | $\mathbf{98.3_{\pm0.1}}$ | $\mathbf{98.7_{\pm0.0}}$ |
| **End + BB** | $77.7_{\pm0.3}$ | $\mathbf{84.7_{\pm1.3}}$ | $88.6_{\pm0.2}$ | $93.6_{\pm0.1}$ | $97.8_{\pm0.0}$ | $98.3_{\pm0.1}$ |

samples to be closer than the different class samples, but it still cannot penalty the same class and same bias samples being closer than the same class but different bias samples. The last loss variant does not use the cross-entropy loss on training the feature representation and counts on the model's ability to find the sweet spot that is expected to be an unbiased feature representation. However, even if the model only learned the target features, the second contrastive loss keeps on giving a penalty to the model since the different bias samples are mostly the different class samples. Therefore, the final performance of the model is worse than BiasCon loss.

### D.3 Joint training of baselines with the BiasBal loss

Our proposed BiasBal loss can replace the cross-entropy loss and it shows high performance on various datasets. The BiasBal loss can also improve the baseline methods that use the cross-entropy loss. To check this, we jointly use the BiasBal loss with EnD [17] and ReBias [1] which are previously the state-of-the-art debiasing methods of the known/unknown bias label cases. Note that ReBias is a method that does not use the bias labels, but here we assume using it on the known bias label case and replace the cross-entropy loss with BiasBal loss in the ReBias training. We evelute the performance of joint use of the BiasBal loss on the BiasedMNIST [1]. For ReBias, we report the test accuracy of the last epoch, and for EnD, we report the test accuracy of the epoch with the highest validation accuracy.

Table A10 shows the unbiased accuracy of the baseline methods trained with the BiasBal loss. For ReBias, joint use of the BiasBal loss improves the performance of the baseline method for all cases. However, BiasBal loss is not helpful for EnD loss in most cases. We suspect that this is due to the hyperparameter settings of the End+BB loss. For End+BB loss, we use the same hyperparameters reported in the original EnD paper [7], which are results of the rigorous hyperparameter search, and we only search for the weight hyperparameter applied to EnD which balances its effect with the BiasBal loss. However, we find that EnD is very sensitive to the hyperparameter settings, and thus we expect that the same hyperparameters do not work when it is jointly used with the BiasBal loss.

### D.4 Ablation study of the BiasCon loss

In order to check whether the effectiveness of our method indeed comes from our bias-contrastive framework, we do the ablation study on the BiasCon loss. We first assess the removal of the augmentation and the multiviewed batch. Upon that, we also remove the oversampling used for the

Table A11: Ablation study of BiasCon loss. We first remove the additional augmentation and multiviewed batch. We then remove the oversampling with $Q(i)$. We report the unbiased accuracy and standard error on the BiasedMNIST [1] dataset with various target-bias correlations.

| Method | 0.999 | 0.997 | 0.995 | 0.99 | 0.95 | 0.9 |
|---|---|---|---|---|---|---|
| BiasCon + BiasBal | $94.0_{\pm 0.6}$ | $\mathbf{97.3_{\pm 0.1}}$ | $\mathbf{97.7_{\pm 0.1}}$ | $\mathbf{98.1_{\pm 0.1}}$ | $\mathbf{98.9_{\pm 0.0}}$ | $\mathbf{99.3_{\pm 0.0}}$ |
| BiasCon | $\mathbf{94.5_{\pm 0.4}}$ | $97.0_{\pm 0.0}$ | $97.4_{\pm 0.1}$ | $97.7_{\pm 0.1}$ | $98.6_{\pm 0.1}$ | $98.9_{\pm 0.1}$ |
| BiasCon − Aug. | $89.4_{\pm 0.5}$ | $93.8_{\pm 0.1}$ | $94.6_{\pm 0.4}$ | $96.3_{\pm 0.2}$ | $98.0_{\pm 0.1}$ | $98.4_{\pm 0.1}$ |
| BiasCon − Aug. − Samp. | $86.6_{\pm 1.2}$ | $95.4_{\pm 0.1}$ | $96.2_{\pm 0.1}$ | $97.6_{\pm 0.0}$ | $97.8_{\pm 0.1}$ | $98.2_{\pm 0.0}$ |
| EnD [17] | $59.5_{\pm 2.3}$ | $82.7_{\pm 0.3}$ | $94.0_{\pm 0.6}$ | $94.8_{\pm 0.3}$ | $98.3_{\pm 0.1}$ | $98.7_{\pm 0.0}$ |

Table A12: The unbiased accuracy (UA), bias-conflict accuracy (BA), equalized odds (EO), and demographic parity (DP) of the model trained on the UTKFace [22] dataset. We compare with an additional fairness baseline LAFTR [12] with target fairness criteria, DP and EO. For better comparison, we scale DP and EO by 100, and report % value.

| Bias | Metric | Vanilla | DI [19] | LAFTR-DP [12] | LAFTR-EO [12] | **BiasCon** | **BiasBal** | **BC+BB** |
|---|---|---|---|---|---|---|---|---|
| Race | UA (↑) | $87.4_{\pm 0.3}$ | $88.9_{\pm 1.2}$ | $82.1_{\pm 0.7}$ | $82.1_{\pm 0.6}$ | $90.3_{\pm 0.2}$ | $90.4_{\pm 0.3}$ | $\mathbf{91.0_{\pm 0.2}}$ |
| | BA (↑) | $79.1_{\pm 0.3}$ | $89.1_{\pm 1.6}$ | $83.3_{\pm 1.1}$ | $83.6_{\pm 0.1}$ | $88.8_{\pm 0.5}$ | $\mathbf{89.9_{\pm 0.6}}$ | $89.2_{\pm 0.1}$ |
| | EO (↓) | $16.5_{\pm 0.3}$ | $1.4_{\pm 0.6}$ | $3.6_{\pm 1.0}$ | $3.0_{\pm 1.3}$ | $3.1_{\pm 0.6}$ | $\mathbf{1.3_{\pm 0.3}}$ | $3.5_{\pm 0.2}$ |
| | DP (↓) | $21.7_{\pm 0.3}$ | $5.1_{\pm 1.0}$ | $2.5_{\pm 1.8}$ | $\mathbf{1.9_{\pm 1.1}}$ | $8.7_{\pm 0.5}$ | $6.5_{\pm 0.4}$ | $9.2_{\pm 0.3}$ |
| Age | UA (↑) | $72.3_{\pm 0.3}$ | $75.6_{\pm 0.8}$ | $67.4_{\pm 0.5}$ | $64.7_{\pm 0.9}$ | $75.7_{\pm 0.2}$ | $78.8_{\pm 0.4}$ | $\mathbf{79.1_{\pm 0.3}}$ |
| | BA (↑) | $46.5_{\pm 0.2}$ | $60.0_{\pm 0.2}$ | $41.2_{\pm 0.5}$ | $43.2_{\pm 2.5}$ | $61.7_{\pm 0.5}$ | $\mathbf{76.7_{\pm 3.2}}$ | $71.7_{\pm 0.8}$ |
| | EO (↓) | $53.8_{\pm 0.2}$ | $33.3_{\pm 1.7}$ | $54.1_{\pm 1.7}$ | $44.9_{\pm 3.0}$ | $29.3_{\pm 0.8}$ | $30.2_{\pm 0.8}$ | $\mathbf{28.1_{\pm 1.1}}$ |
| | DP (↓) | $54.8_{\pm 0.2}$ | $34.5_{\pm 1.4}$ | $54.8_{\pm 1.8}$ | $45.2_{\pm 2.9}$ | $31.5_{\pm 0.7}$ | $\mathbf{9.8_{\pm 3.9}}$ | $18.7_{\pm 1.3}$ |

batch sampling of the BiasCon loss, which is solely applying the contrastive loss. We evaluate the ablations of the BiasCon loss components on the BiasedMNIST [1] dataset.

We report the results in Table A11. As explained in Section A.2, the contrastive loss (row named BiasCon − Aug. − Samp.) itself is very effective, outperforming the previous state-of-the-art in most cases. Upon that, the oversampling strategy (row named BiasCon − Aug.) is critical for the case with a high target-bias correlation ($\rho = 0.999$), but the gain decreases or may adversely impact when the correlation is not very high. These results are expected because when the correlation decreases, a single batch contains enough positive pairs, diminishing the gain of oversampling. Moreover, the adverse impact that oversampling can bring is that the model focuses on learning features that are only useful for the subset of the dataset and fails to learn more general features. If we utilize the augmentation and multiviewed batch (our proposed BiasCon loss), it remedies the shortcomings of the oversampling as it offers more diverse views of the oversampled data, and thus the model can learn more useful and generalized invariances.

### D.5 Comparison with fairness baselines

Learning fair representation [3, 12, 21] is highly related topic to the model debiasing problem. These works define various criteria for group fairness of the model prediction. Two representative fairness criteria are the following [20].

- Demographic parity (DP): The classifier prediction $\hat{Y}$ satisfies the demographic parity with respect to sensitive attributes $S$ if $\hat{Y} \perp\!\!\!\perp S$. The metric to measure the quality of the demographic parity is:

$$\Delta DP := |P(\hat{Y} = 1|S = 1) - P(\hat{Y} = 1|S = 0)|. \tag{D.1}$$

- Equalized odds (EO): The classifier prediction $\hat{Y}$ satisfies the demographic parity with respect to sensitive attributes $S$ if $(\hat{Y} \perp\!\!\!\perp S) \mid Y$. The corresponding metric is:

$$\Delta EO := \mathbb{E}_y \Big[ |P(\hat{Y} = 1|S = 1, Y = y) - P(\hat{Y} = 1|S = 0, Y = y)| \Big]. \tag{D.2}$$

To measure the fairness of the prediction of the debiased model, we measure DP and EO of our proposed methods and baselines on UTKFace [22] dataset. We also compare with LAFTR [12], a baseline method for learning fair representation. LAFTR proposes an adversarial training scheme which directly optimizes the fairness criterion (DP or EO). Note that we omit the decoder loss of LAFTR as it requires a large additional decoder model to reconstruct an image. The results in Table A12 show that our proposed BiasCon and BiasBal losses fairly induce the fair representation learning, but unlike the results on the accuracy measures, they do not outperform the other baselines in the *Race* task. LAFTR-DP and LAFTR-EO show good EO and DP results in the *Race* task, but both methods result in worse unbiased accuracy than the vanilla model. This implies the trade-off between accuracy and the fairness criteria. Finally, different trends of EO, DP, and two accuracy measures indicate the importance of the evaluation with diverse metrics.