# OpenReview forum: "Unbiased Classification through Bias-Contrastive and Bias-Balanced Learning"
_NeurIPS.cc/2021/Conference — NeurIPS 2021 Poster_

### Official Review · Reviewer_YM7p · 2021-06-26

**Rating:** 8
**Confidence:** 4

**Summary:**

The authors propose three debiasing methods. First, Bias-Contrastive Loss intends to bring the feature representations for samples with the same target class but different bias labels together. Second, the authors derive a Bias-Balanced Loss, which optimizes for unbiased distribution i.e., a distribution where target class and bias labels have uniform correlations. Third, they propose Soften Bias-Contrastive loss, which does not require bias labels, but instead uses a bias-capturing model to modulate the contrastive loss. The experiments are conducted both with and without bias labels and show improvements on all of the benchmarks.

**Limitations And Societal Impact:**

While the authors mention not testing on more realistic benchmarks e.g., VQA, it might be worthwhile to also add that in such cases, biases may stem from multiple sources and it may not be trivial to label all of them or design bias-capturing models for all of them (which are required by the proposed methods).

**Main Review:**

Overall, the proposed techniques are novel and the experiments are well set up to demonstrate their efficacies. Building bias-resistant techniques based on the recently rising self-supervised methods is a valuable contribution to the field. Both BiasCon and BiasBal are well motivated and show improvements over other techniques that rely on bias labels. The SoftCon loss also improves upon other techniques that do not require bias labels. For this, SoftCon uses feature similarity/distance from bias-capturing models, which is different from how the previous methods e.g., LfF used them. Overall, the results are impressive, the techniques will be useful and the paper is very well written.

I will now move on to clarification questions, addressing which may help improve the paper:

[Q1] The very first question I had with the contrastive loss was: did the multi view batches already encode invariances? E.g., using grayscale for BiasedMNIST would be a trivial solution. While the authors mention the exact augmentation strategies in the supplementary section, it might be worthwhile to have at least a sentence stating whether or not the creation of the multi-view batch itself would help learn the signal in the main text itself.

[Q2] For the Biased Contrastive Loss, it wasn’t clear why the samples with the same class, same bias were used as negatives. Intuitively, all instances from the same class should have similar representations, so should be used as positive pairs. What happens if the same class, same bias samples are used as positives instead? Which one is better and why?

[Q3] While the oversampling ablation is mentioned in the supplementary, it might be worthwhile to have resampling/loss reweighting techniques as comparison methods for all the datasets.
While simple, loss reweighting can mitigate biases under sufficient regularization [1], and thus might have practical utility. Also, the intuition behind the simple, classical techniques of resampling/reweighting based on target and bias labels is the same as the goal of Theorem 1 i.e., to optimize for uniform correlation between target and bias labels. As such, both theoretical and empirical comparisons with them would be a huge plus. It would be particularly interesting to have theoretical analyses supported by empirical results as to why/if BiasBal loss is better than the classical techniques. What is lacking in these classical techniques, that are otherwise very intuitive?
At the very least, [1] should be cited here.

[Q4] The experiments section would be more insightful if the authors ran SoftBiasCon on datasets where it is not obvious how one can develop a bias-capturing model. The two datasets tested for SoftBiasCon had similar bias-encoding models (geared towards capturing background color/texture), but it is not obvious how an architecture could be developed for capturing “gender” bias in CelebA. Of course, perhaps the bias amplified model from LfF could be used for this case. In any case, it would be nice to have SoftBiasCon loss be run on all datasets, since bias mitigation without bias labels would be highly desirable in practice.

[Q5] While the bias labels were not used during training for the SoftCon loss, were they used for model selection/hyperparameter tuning? It would be good to explicitly mention this.

[1] Sagawa, Shiori, et al. "An investigation of why overparameterization exacerbates spurious correlations." International Conference on Machine Learning. PMLR, 2020.

**Time Spent Reviewing:**

5

---

> ### Author Response · Authors · 2021-08-10
> **Response to Reviewer YM7p**
>
> We sincerely thank the reviewer for the constructive feedback.
> We tried our best to address the reviewer’s concerns and questions. Hope this response answers the reviewer’s questions.
>
> ---------
> **[Q1. Did the multiviewed batches already encode the invariances?]**
>
> Thank you for asking about this issue. We want to clarify that as explicitly explained in Appendix C, we used the standard augmentation set and did not selectively choose an augmentation that can remove the bias factor (e.g., grayscale augmentation for BiasedMNIST).
> Nevertheless, as the reviewer pointed out, some may think that the debiasing ability comes from the augmentation itself.
> To suppress such concern, in Appendix D.4 and Table A15, we did an ablation study on the augmentation used to generate multiviewed batch in the BiasCon loss.
> For convenience, we copied the values in Table A15:
>
> ||0.999|0.997|0.995|0.99|
> |:----:|:----:|:----:|:----:|:----:|
> |BiasCon|92.0|96.8|97.3|98.0|
> |BiasCon w/o Aug|89.9|95.6|96.6|97.5|
>
> The results show that there is a slight drop in the performance when we remove the multiviewed batch, but the performance is still high compared to the previous baselines.
> Therefore, we can conclude that the bias-contrastive framework itself is an effective method for debiasing the model, and the multiviewed batch gives an extra performance gain.
> We speculate that additional performance gain of the multiviewed batch is originated from the well-known observation that strong data augmentation is effective in the contrastive framework [a].
>
> Meanwhile, there exists a line of work that uses strong data augmentation for debiasing, such as shape-texture mixing for removing the texture biases [b].
> Combining a useful augmentation strategy for debiasing with our bias-contrastive framework would be interesting future work.
>
> ---------
> **[Q2. Design choice of the BiasCon loss]**
>
> The fundamental issue of learning with bias is that learning through the standard cross-entropy (CE) loss learns bias faster than signal. Therefore, in designing our BiasCon loss, we focus more on preventing the classifier from using bias information rather than grouping the same class together (The latter, we believe, will be handled well by CE loss). From this point of view, we focus more on cases belonging to the same class but with different biases. Compared to considering all instances belonging to the same class regardless of bias, this approach can be seen as a focus on harder positives. Indeed recent studies have shown that considering hard positives can be advantageous in contrastive learning. [c,d]
>
> Nevertheless, we also think it is important to empirically compare the performance of the variant that the reviewer proposed:
>
> BiasCon-V2: positive pair - same class / negative pair - different class regardless of bias (same as SupCon loss, but trained in joint with CE loss), without oversampling
>
> BiasCon-V2-ovs: BiasCon-V2 with the same oversampling strategy as BiasCon
>
> ||0.999|0.997|0.995|0.99|
> |:----:|:----:|:----:|:----:|:----:|
> |BiasCon-V2|52.5|75.0|84.4|90.8|
> |BiasCon-V2-ovs|65.8|85.8|91.7|95.0|
> |BiasCon|92.0|96.8|97.3|98.0|
>
>
> We observe that using the same class but different bias samples as a positive pair is better at preventing the model from falling into shortcuts using bias features.
> We think this experiment is crucial to justify the design choice of our BiasCon loss. Thank you for the feedback.
>
> ---------
> **[Q3. Comparison with loss reweighting method]**
>
> Thank you for the valuable suggestion. We agree that both loss reweighting and our BiasBal loss both target to optimize the model toward uncorrelated data distribution.
> The difference is in how the two methods induce the model to optimize toward the uncorrelated data distribution.
> The loss reweighting achieves this by applying the importance sampling strategy on the risk minimization framework, i.e., $L= \mathbb{E}[w \cdot l(x, y)]$, where $w=\frac{1}{p(y|b)}$.
> On the other hand, our BiasBal achieves this by modifying the logits so that the original model prediction $p_{train}(y|x,b)$ become $p_u(y|x,b)$.
>
> To empirically measure the performance of two methods, we conducted experiments on the BiasedMNIST and UTKFace datasets. RW stands for the loss reweighting method. Results of BiasBal and BiasCon are copied from Table 1 and 3 of our paper.
>
>
>
> **BiasedMNIST**
>
> ||0.999|0.997|0.995|0.99|
> |:----:|:----:|:----:|:----:|:----:|
> |RW|79.2|93.1|94.5|96.5|
> |BiasBal|76.8|92.0|94.4|96.6|
> |BiasCon|92.0|96.8|97.3|98.0|
>
>
> **UTKFace**
>
> |Task|Acc. Type|RW|BiasBal|BiasCon|
> |:----:|:----:|:----:|:----:|:----:|
> |Race|Unbiased|89.1|89.3|90.4|
> ||Bias-conflict|87.1|88.3|88.9|
> |Age|Unbiased|79.4|78.3|79.0|
> ||Bias-conflict|74.3|72.1|75.0|
>
>
>
> The result shows that the two methods show comparable performance throughout the dataset.
> This is expected as both methods induce the same objective to the model.
> Both the loss reweighting and BiasBal are simple baselines that recent papers did not explicitly compare.
> We will add a comparison with RW and add the reference [f] the reviewer suggested.
>
> ---------
> **[Q4. Performance of the SoftCon loss on the dataset where designing a bias-capturing model is infeasible ]**
>
> We agree that designing a bias-capturing model may be difficult when the bias is in the form of sensitive attributes such as race or gender.
> In such cases, as the reviewer suggested, we may still use the property that bias features are learned much faster than the target features, as in [e].
> Specifically, we can train a bias-capturing model by emphasizing samples with small loss using generalized cross-entropy (GCE) loss.
> This way, without explicit information about the bias, the model will be more biased toward such attributes.
> In order to check the feasibility of this method, we made some preliminary experiments on the CelebA and UTKFace datasets.
> For fast comparison, results of vanilla training with CE loss, EnD[g], and BiasCon are directly copied from Table 2 and 3 of our paper. Please note that EnD and BiasCon losses are not directly comparable since they use bias labels.
>
> **CelebA**
>
> |Task|Acc. Type|SoftCon|Vanilla|EnD[g]|BiasCon|
> |:----:|:----:|:----:|:----:|:----:|:----:|
> |Blonde|Unbiased|84.1|82.6|88.0|89.5|
> ||Bias-conflict|74.4|66.4|78.7|84.6|
> |Makeup|Unbiased|77.4|76.5|77.8|79.8|
> ||Bias-conflict|61.0|57.1|62.0|65.2|
>
> **UTKFace**
>
> |Task|Acc. Type|SoftCon|Vanilla|EnD[g]|BiasCon|
> |:----:|:----:|:----:|:----:|:----:|:----:|
> |Race|Unbiased|87.0|87.2|88.1| 90.4|
> ||Bias-conflict|80.2|79.1|81.4|88.9|
> |Age|Unbiased|74.6|72.4|74.9|79.0|
> ||Bias-conflict|59.2|47.5|63.2|75.0|
>
> The results show that inducing the bias-capturing model trained with GCE loss can give noisy but meaningful soft labels of positive pairs with shared bias that SoftCon loss can utilize.
> We believe it can be a good future research direction to deal with this issue in more depth and find a fundamental solution.
>
>
>
> ---------
> **[Q5. On the hyperparameter choice of the SoftCon loss]**
>
> We used the bias labels or texture group in the test set only to calculate the unbiased accuracy uniformly for all models including our model. We will clarify this in the final version.
>
> ---------
> [a] Chen et al., A simple framework for contrastive learning of visual representations, 2020
>
> [b] Li et al., Shape-texture debiased neural network training, 2021
>
> [c] Khosla et al., Supervised Contrastive Learning, 2020
>
> [d] Zhang et al., Self-supervised representation learning via adaptive hard-positive mining, 2021
>
> [e] Nam et al., Learning from failure:Training debiased classifier from biased classifier, 2020
>
> [f] Sagawa et al., "An investigation of why overparameterization exacerbates spurious correlations." 2020.
>
> [g] Tartaglione et al., End: Entangling and disentangling deep representations for bias correction, 2021

---

> > ### Comment · Reviewer_YM7p · 2021-08-13
> > **Recommending for acceptance**
> >
> > I would like to thank the authors for their detailed replies. They have satisfactorily addressed all of my concerns.
> >
> > 1. The same class samples being treated as negative pairs (when they have the same bias) was initially confusing, but the provided results do show benefits over treating them as positives.
> >
> > 2. The authors have satisfactorily answered my questions about whether augmentations contained invariances already. I would still recommend mentioning this briefly in the main text itself, so the reader does not have to go through the appendix to verify this.
> >
> > 3. The comparisons against classical methods of re-weighting will be useful. I appreciate the authors running this.
> >
> > 4. I agree with bHbH’s concerns about the connections between losses not being tight enough. Having said that, the losses individually themselves are good contributions to the field, so I am recommending this paper for acceptance.

---

> > > ### Author Response · Authors · 2021-08-15
> > > **Further Response to Reviewer YM7p**
> > >
> > > We thank the reviewer again for the feedback. We will add the additional experiments and discussion you mentioned in the final version.

---

### Official Review · Reviewer_bHbH · 2021-07-06

**Rating:** 7
**Confidence:** 4

**Summary:**

This paper proposed several debiasing losses for unbiased image classification. First, this paper proposed two losses for known bias labels. motivated by contrastive learning, the Bias-Contrastive (BiasCon) loss for better representation learning. The Bias-Balanced (BiasBal) regression is proposed for highly imbalanced target-bias correlation. Second, the Soften Bias-Contrastive (SoftCon) is proposed for unknown bias labels. Experiments on BiasedMNIST, CelebA and UTKFace show that the proposed debiasing method outperforms state-of-the-art methods by a large margin whenever the bias label is available.

**Limitations And Societal Impact:**

The authors have adequately addressed the limitations and potential negative societal impact of their work.

**Main Review:**

Strengths:

+ This paper reveals the potential and effectiveness of contrastive learning in debiased image classification. The contrastive-learning-based loss is simple but very effective.

+ Experimental results on several benchmark datasets show that the proposed methods outperform state-of-the-art debiasing methods by a large margin with or without bias labels.

+ Ablation studies are sufficient to show the effectiveness of each component.

Weaknesses & Questions:

- The connection between BiasCon and BiasBal losses are not tight enough. It seems that these two losses are like a bag of tricks rather than a whole. Also, BiasBal is not used without bias labels. Even without BiasBal, I still think this paper is good. However, BiasBal seems a little incompatible in this paper. Perhaps a better introduction of BiasBal would help with this.

- Another question on BiasBal. While BiasBal achieves competitive results in Tables 2 and 3, the performance in Table 1 is much lower than BiasCon when the bias and label are highly correlated. Recall that BiasBal is proposed to tackle the issue of a highly imbalanced correlation between label and bias. Is there any explanation for this?

- The performances of Vanilla models with bias labels were implemented (Table 1-3) while those without bias labels were borrowed from ReBias (Table 4, 5). I wonder (1) why the vanilla models were not implemented in Tables 4 and 5, (2) whether the increase comes from the implementation. Results of vanilla models obtained by the same reimplementation would answer these questions.

- It is not clear how Eq. (5) and (6) are theoretically correct. In Eq. (5) and (6), why can we replace $P_{u}$ with $P_{train}$?

- After reading the paper, I am supervised by the success of contrastive learning in debiased image classification, but still not clear why contrastive learning can outperform others by a large margin (e.g., 93.3 vs. 52.3 on BiasedMNIST with corr=0.999 in Table 1). Is there a more intuitive or convincing explanation for this increase?

- As shown in Tables A4 and A6 in the appendix, the hyperparameters $\alpha$ are various for different correlations between label and bias. Is it a fair experiment setting? Or, is the hyper-parameter determined based on the test set? It is okay if this is the commonly used setting in other related works. In my opinion, the hyperparameters should be the same for different correlations as the correlations are supposed to be unknown in real-world applications.


==================== After rebuttal ======================

According to the authors' feedback, most of my major concerns have been addressed, and I am happy to vote for acceptance.

The strengths and contributions of this paper are:

(1) Two losses for known bias labels and one variant for unknown bias labels for unbiased image classification. BiasCon is proposed for better representation learning motivated by contrastive learning, and BiasBal is proposed for highly imbalanced target-bias correlation. A variant of BiasCon, SoftCon, is proposed for unknown bias labels. These losses are simple but effective for unbiased image classification.

(2) The paper is well written and easy to follow. The contributions are clearly stated.

(3) Experiments and ablation studies are well designed and demonstrate the effectiveness of the proposed losses.

My major concerns were:

(1) The connection between BiasCon and BiasBal losses is not tight enough. According to the authors' feedback, I acknowledge the contribution of each loss to unbiased image classification. However, it would be greater if these two losses could be unified into one framework rather than two independent techniques. Due to this major concern, I would not give a higher score.

(2) The reason why contrastive learning works. Reviewer bmYk and YM7p raise a similar concern via a more detailed question: how and why the choice of BiasCon is determined, especially positive and negative pairs. The authors' feedback provides more experimental results to empirically compare BiasCon and other variants, which I think answers the "how" question. For the "why" question, I have left a post-rebuttal note to request the authors a more comprehensive and conclusive explanation.

**Time Spent Reviewing:**

4

---

> ### Author Response · Authors · 2021-08-10
> **Response to Reviewer bHbH**
>
> We sincerely thank the reviewer for the constructive feedback.
> We tried our best to address the reviewer’s concerns and questions. Hope this response answers the reviewer’s questions.
>
> ---------
> **[Q1. Connection between BiasCon and BiasBal]**
>
> We agree that the connection between the proposed BiasCon and BiasBal losses is not tight enough. As the reviewer pointed out, two losses can be seen as a bag of tricks that we can apply when the bias label is available.
> However, we would like to emphasize the strength of the BiasBal loss. The BiasBal loss is very easy to implement, does not contain any hyperparameters, and most of all, it shows good performance in most of the experiments. For example, in the unbiased image classification task on the CelebA dataset where Blonde is the target attribute (Table 2), BiasBal shows even better performance than BiasCon. It also gives additional performance gain when it is jointly trained with BiasCon loss. Therefore, the purpose of proposing BiasBal loss is to give a simple but powerful and versatile loss that can replace the cross-entropy (CE) loss.
>
> ---------
> **[Q2. Performance of the BiasBal loss when the correlation is very high]**
>
> The basic assumption for the BiasBal loss is that $P(X|Y, B)$ remains the same for train and test data distribution. However, when the correlation is very high, there are only a few minor class (under given bias) samples.
> Therefore, when the samples for a given Y, B pair are very scarce, we cannot assume that $P(X|Y, B)$ remains the same.
> For example, in the BiasedMNIST dataset with a correlation of 0.999, under bias class id 2 (samples with blue background), the number of samples for each target class are [1, 1, 5952, 1, 1, 0, 1, 1, 1, 1].
> Note that for a joint pair $(Y, B)$ that has no sample (which appears only when the correlation is 0.999), we added $\epsilon=1e-9$ instead of $p(y|b)=0$ in the calculation of BiasBal loss.
> In gist, we conjecture that the performance degradation of the BiasBal loss on a dataset with extremely high correlation is because the underlying assumption of the BiasBal loss is not satisfied. We will add this discussion in the revision.
>
>
>
>
> ---------
> **[Q3. Performance of the Vanilla model]**
>
> We borrowed the results of vanilla model on BiasedMNIST (Table 1 and 4) and ImageNet-9 (Table 5) from [a] while we implemented the results on CelebA (Table 2) and UTKFace (Table 3) as these are experiments newly designed by ourselves (We apologize for missing dagger symbol in Table 1).
> To alleviate the concern that the performance increment is due to implementation difference, we measure the performance of our reimplementation of the vanilla model for BiasedMNIST, as the reviewer suggested:
>
> ||0.999|0.997|0.995|0.99|
> |:----:|:----:|:----:|:----:|:----:|
> |Vanilla|12.0|61.0|76.8|88.3|
>
> Most results are similar to the one reported in [a] but there is a significant difference in the average unbiased accuracy for correlation 0.997.
> We checked that the standard deviation of the accuracy reported in ReBias [a] is 21.0, which is very high, considering that the mean accuracy is 33.4. Our implementation also had a relatively high standard deviation of 4.74.
> The large difference might come from the high instability of the model performance trained with CE loss on the correlation 0.997.
> We will replace the Vanilla model performance with our implementation in Table 1 and 4.
> We can expect similar results for Table 5, and we will replace them all with the results of our implementation in the final version.
>
>
> ---------
> **[Q4. On the definition of $P_u$ using $P_{train}$ (Equation 5 and 6)]**
>
> We apologize for the confusion. We want to clarify that Equation 5 and 6 are defining the uncorrelated data distribution $P_u$ which has the same data generation process with training data distribution (i.e., $P_{train}(X|Y, B) = P_u(X|Y, B)$, but the correlation between Y and B is changed to become uniform (i.e., $P_u(Y|B) = 1/C$). That is, we are not just replacing terms, but we are defining a distribution that does so by construction.
> We think that the reader can understand more easily if we first define $P_u(Y, B)$ and then define $P_u(X, Y, B)$.
> We will add a clearer explanation about the definition of $P_u$ in the final version.
>
> ---------
> **[Q5. Effectiveness of contrastive learning in debiasing]**
>
> To be honest, we were also very surprised by the effectiveness of contrastive learning in debiasing. Despite a lot of effort after seeing the results, we couldn’t find a special reason for such a big improvement with contrastive learning. We leave it as a future research topic to seek a deeper understanding of this phenomenon and to broaden our understanding on tasks beyond debiasing in which the contrastive learning can work very effectively.
>
>
> ---------
> **[Q6. Hyperparameter choice]**
>
> Following the strategy in related works, we chose the best hyperparameter set that gives the highest unbiased accuracy with a naive grid search.
> Since we applied this strategy for all methods, we can compare the best performance of the proposed method and baselines.
> In the real-world setting, we believe that if we can construct a small test set with possible sensitive attributes that the model might be biased toward, we can find the best settings that the model can achieve.
>
> ---------
> [a] Bahng et al., Learning De-biased Representations with Biased Representations, 2020

---

> > ### Comment · Reviewer_bHbH · 2021-08-13
> > **A further question/suggestion on Q5**
> >
> > Thanks for the authors' comprehensive feedback! I have noticed that Reviewer bmYk and YM7p also asked about the choice of the BiasCon loss, and the experimental results in the feedback empirically verified the effectiveness of the BiasCon loss. Could the authors summarize these added results and give a more comprehensive and conclusive reason for the design choice? This may address my Q5 better. Thank you!

---

> > > ### Author Response · Authors · 2021-08-15
> > > **Further Response to Reviewer bHbH**
> > >
> > > Thank you for the further suggestion. By answering the questions of the Reviewer bmYk and YM7p, we could further empirically justify the design choice of the BiasCon loss. Here, we summarize the experimental results of alternative design choices.
> > >
> > > ---------
> > > **[Q1. Design choice of the BiasCon loss]**
> > >
> > > The fundamental issue of learning with bias is that learning through the standard cross-entropy (CE) loss learns bias faster than signal. Therefore, in designing our BiasCon loss, we focus more on preventing the classifier from using bias information rather than focus on grouping the same class together (The latter, we believe, will be handled well by CE loss). From this point of view, we focus more on cases belonging to the same class but with different biases. Compared to considering all instances belonging to the same class regardless of bias, this approach can be seen as a focus on harder positives. Indeed recent studies have shown that considering hard positives can be advantageous in contrastive learning. [a,b]
> > >
> > > To further justify the design choice of the BiasCon loss, we empirically compared the variants that the Reviewer bmYk and YM7p proposed.
> > >
> > > We first recapitulate the positive and negative pairs used for our BiasCon loss.
> > >
> > > BiasCon (ours): positive pair - same class but different bias / negative pair - others (different class + same class and same bias)
> > >
> > > Reviewer bmYk and YM7p asked to compare with the variant that includes all the same class samples as a positive pair, as those samples share the same target features. (Variant 1, 2)
> > >
> > > Variant 1: positive pair - same class / negative pair - different class but same bias
> > >
> > > Variant 2: positive pair - same class / negative pair - different class regardless of bias (same as SupCon loss, but trained in joint with CE loss)
> > >
> > > Variant 2-ovs: Variant 2 with the same oversampling strategy as BiasCon
> > >
> > > Reviewer bmYk further asked to compare with the variant that removes same class and same bias samples from the negative pairs, as having them as a negative samples can harm the effect of the cross-entropy (CE) loss. (Variant 3)
> > >
> > > Variant 3: positive pair - same class but different bias / negative pair - different class, with the same oversampling strategy as BiasCon
> > >
> > > ||0.999|0.997|0.995|0.99|
> > > |:----:|:----:|:----:|:----:|:----:|
> > > |Variant 1|23.0|67.7|75.9|89.1  |
> > > |Variant 2|52.5|75.0|84.4|90.8|
> > > |Variant 2-ovs|65.8|85.8|91.7|95.0|
> > > |Variant 3|83.0|87.7|88.5|89.4|
> > > |BiasCon|92.0|96.8|97.3|98.0|
> > >
> > >
> > > By comparing the results of Variant 2-ovs and Variant 3, we can find that using the same class but different bias samples as positive pairs as in our BiasCon loss is effective for preventing the model from falling into shortcuts using bias features.
> > > Moreover, by comparing Variant 3 and BiasCon, giving a penalty to samples with the same class and same bias samples is also important.
> > >
> > > Through this experiment, we can justify the effectiveness of our BiasCon loss.
> > >
> > > ---------
> > > [a] Khosla et al., Supervised Contrastive Learning, 2020
> > >
> > > [b] Zhang et al., Self-supervised representation learning via adaptive hard-positive mining, 2021

---

> > > > ### Comment · Reviewer_bHbH · 2021-08-24
> > > > **Acknowledgment of the response**
> > > >
> > > > Sorry for the late acknowledgment. Thank the authors for their comprehensive response. This has addressed my concern and I am happy to accept this paper.

---

### Official Review · Reviewer_Eyr1 · 2021-07-17

**Rating:** 6
**Confidence:** 4

**Summary:**

The paper propose two method, BiasCon and BiasBal, to mitigate the dataset bias. In particular, BiasCon is a contrastive-like objective that pulls a pair that shares the same target class but different bias feature. BiasBal objective is derived via assuming the data generating process. The authors also propose SoftCon, an extension of BiasCon, to tackle the unknown bias label. The proposed methods are evaluated on BiasedMNIST, CelebA and UTKFace dataset.

# POST-REBUTTAL
The additional results look good. After reading these and my co-reviewers comments, I still advocate for weak accept (score 6).

**Limitations And Societal Impact:**

The authors adequately addressed the limitations and potential negative societal impact.

**Main Review:**

Overall, the paper address an important problem in machine learning. The paper is well-written and the contributions are stated clearly. I have a few questions and concerns.

Positive Points:

1. The proposed BiasBal is interesting. It seems that it is a simple yet effective objective for tacking the dataset bias.

2. The proposed approach can also be partially extended to the case when the bias label is missing. It could be very influential since in real world application such as medical system, the bias label (e.g., gender) is not accessible.


Negative Points:

1. The paper does not discuss or compare to existing works about fairness. The bias variable is essentially the sensitive attributes in fairness literatures, e.g., the gender attribute in CelebA and UTKFace experiments. There are existing works [1,2,3] that leverage the sensitive attribute to learn fair representations or build fair classifier.

2. Besides machine learning fairness literatures, there is another line of works that solves similar problem under the domain generalization or casual inference framework. I encourage authors to include some standard baselines such as Invariant Risk Minimization [4] or Domain-invariant Networks [5]? In particular, these works treat the domain-specific features as biased features. Therefore, the variable B could be domain index.

3. The main metric used in the paper is the unbiased accuracy. It would be great to measure and discuss the performance w.r.t. demographic parity or equalized odds on CelebA and UTKFace dataset. To ensure machine learning fairness, unbiased accuracy only provides little informations.

Other comments:

1. I suggest moving the graphical model in appendix to section 3.3. This would help readers understand equation (5) and (6) more.

2. It would be great to perform some analyses to the BiasCon objective. For instance, maybe one can link this to the mutual information between the feature and feature bias B?

3. The notation is a bit confusing. It seems that $p$ has been used to denote several distributions.

Overall, I think this paper address an important problem that will be of interest to a wide audience in NeurIPS. The method is stated clear and the experiments are sound. Nevertheless, there is lack of discussions and comparison to previous works. The paper would be much more polished if the authors can provide more empirical comparison to other baselines and metrics.


[1] Zemel et al., Learning Fair Representations, 2013

[2] Madras et al., Learning Adversarially Fair and Transferable Representations, 2018

[3] Creager et al., Flexibly fair representation learning by disentanglement, 2019

[4] Arjovsky et al., Invariant Risk Minimization, 2020

[5] Ganin et al, Domain-Adversarial Training of Neural Networks, 2015

[6] Li et al., Information-Preserving Contrastive Learning for Self-Supervised Representations, 2021


**Time Spent Reviewing:**

4

---

> ### Author Response · Authors · 2021-08-10
> **Response to Reviewer Eyr1**
>
> We sincerely thank the reviewer for the constructive feedback.
> We tried our best to address the reviewer’s concerns and questions. Hope this response answers the reviewer’s questions.
>
> ---------
> **[Q1. Comparison with works on fairness]**
>
> While it is common that debiasing papers such as [d,e,f] do not explicitly deal with their research in terms of fairness, we also agree with the reviewer that it would be instructive to compare against fairness baselines [a,b,c] as both works target similar objectives. In order to bridge between two similar tasks, we performed some preliminary experiments to compare our method with fairness baselines.
> Among the baselines the reviewer suggested, we implemented LAFTR [b] as it can be seamlessly applied to the model we used for the unbiased image classification of the UTKFace dataset.
> Since the original LAFTR [b] is tested on tabular datasets, we only borrowed the concept of adversarial training with fairness objective loss and trained the same ResNet18 model, following the setup of our experiments. We did not use the decoder loss, as it requires a large decoder model to reconstruct an image.
> We tested two versions of LAFTR, the LAFTR-DP and LAFTR-EO, which use demographic parity (DP) and equalized odds (EO) adversarial objective, respectively.
> For fast comparison, results of vanilla training with CE loss, BiasCon, and BiasBal are directly copied from Table 3 of our paper. (For more evaluation results with fairness metrics DP and EO, please refer to the response to Q3 below.)
>
> |Task|Acc. Type|LAFTR-DP|LAFTR-EO|Vanilla|BiasCon|BiasBal|
> |:----|:----:|:----:|:----:|:----:|:----:|:----:|
> |Race|Unbiased|88.8|88.6|87.2|90.4|89.3|
> ||Bias-conflict|83.3|87.4|79.1|88.9|88.3|
> |Age|Unbiased|73.2|74.4|72.4|79.0|78.3|
> ||Bias-conflict|48.6|52.2|47.5|75.0|72.1|
>
> The result implies that fairness baselines are indeed important ones that are comparable to previous works.
> In these preliminary experiments, our BiasCon and BiasBal losses are better than LAFTR, but there may exist room for LAFTR to be improved, such as utilizing additional decoder loss or applying recent techniques used in generative models.
>
> We will add a detailed comparison with the suggested fairness baselines [a,b,c] in the final version of our paper.
>
> ---------
> **[Q2. Comparison with domain generalization or causal inference framework]**
>
> Thank you for the valuable suggestion. We agree that papers on domain generalization [g,h] are also important baselines that recent debiasing papers did not explicitly discuss or compare against in detail [d,e,f].
> However, we would like to emphasize the fact that the method ‘Learning Not to Learn’ (LNL) [i], which we used as a baseline in Section 4.1, already applied and extended the learning objectives and techniques of DANN [h] for the debiasing model prediction task. LNL uses a bias prediction branch with a gradient reversal layer and proposes to jointly optimize the bias prediction loss and target prediction loss as the minimax optimization. We believe that the superiority of our method compared to DANN [h] mentioned by the reviewer (via LNL, its debiasing model version) can be confirmed from the experimental results on various datasets that we have already included in the paper: Biased MNIST (Table 1), CelebA (Table 2), and UTKFace (Table 3).
>
> We will add detailed explanations about the relation between domain generalization [g, h] and debiasing methods including ours in the related work or discussion section in the final version.
>
> ---------
> **[Q3. Additional metrics to evaluate our method]**
>
> We agree that evaluating our method with metrics in fairness literature can give more information about the debiasing performance of our method.
> Toward this, we evaluated our method and baselines with two additional metrics on the UTKFace dataset, as the reviewer suggested: demographic parity (DP) and equalized odds (EO).
>
> |Task|Metric|LAFTR-DP|LAFTR-EO|Vanilla|EnD|DI|LNL|BiasCon|BiasBal|
> |:----|:----:|:----:|:----:|:----:|:----:|:----:|:----:|:----:|:----:|
> |Race|DP|0.142|0.042|0.225|0.185|0.051|0.220|0.076|0.074|
> ||EO|0.097|0.005|0.180|0.140|0.008|0.175|0.028|0.027|
> |Age|DP|0.489|0.368|0.529|0.098|0.450|0.495|0.114|0.047|
> ||EO|0.472|0.345|0.514|0.138|0.432|0.478|0.181|0.244|
>
> As expected, we could find the rough trend that when the model exhibits high unbiased and bias-conflict accuracies, it also achieves better fairness properties in terms of DP and EO.
> However, as the reviewer pointed out, we could perform fine-grained analysis with DP and EO.
> The most noticeable finding from this fine-grained analysis is that in the `race` task, although BiasCon and BiasBal show higher unbiased accuracy than LAFTR-EO, LAFTR-EO shows better DP and EO. This implies that the prediction of the model trained with our BiasCon or BiasBal loss is correlated with the bias factor to some degree, compared to the model trained with LAFTR-EO.
> Still, as our BiasCon and BiasBal loss shows higher unbiased accuracy, they induce the model to learn more discriminative features than LAFTR-EO.
> Combining the strength of two methods for training fair but strong models will be interesting future work.
>
> As a concluding remark, we think analyzing the model with fairness metrics such as DP or EO is crucial to understand the behavior of the model. We will add this analysis in the final version of our paper.
>
> ---------
> **[Q4. Probabilistic interpretation of the BiasCon loss with the mutual information]**
>
> As the reviewer suggested, we can give a probabilistic interpretation of the BiasCon loss using the fact that the contrastive loss follows the formulation of InfoNCE loss [j], and minimizing the InfoNCE loss maximizes a lower bound of mutual information.
> Here, we give a rough sketch of the probabilistic interpretation.
> Let $u_i$ be the random image sampled from the data distribution and $v_i$ be the image from the same data distribution sampled to have the same target class with $u_i$ but with different biases. Thus, $v_i$ is conditioned on $u_i$.
> Let $f$ be the feature extractor of the model.
> Then, with the set of samples $(u_i, v_i), i=1...N$, the BiasCon loss can be approximated as the InfoNCE estimator:
> $$
> L_{\text{BiasCon}}  \approx -\frac{1}{N} \sum\limits_{i=1}^{N} \log \frac{ h(u_i, v_i) }{ \frac{1}{N} \sum_j h(u_i, v_j)}
> $$
>
> where $h(u, v)=\sum_{k,l \in \{ 1,2 \}} \exp{(z_{u,k} \cdot z_{v,l} / \tau)}$, $(z_{u,1}, z_{u,2})$ is a pair of normalized features (i.e., $z(u)=\frac{f(u)}{||f(u)||}$) of augmented view of $u$, and $(z_{v,1}, z_{v,2})$ is defined similarly.
> Thus, we can interpret that the BiasCon loss is inducing the model to maximize the mutual information $I(f(u),f(v))$, which are features of samples from the same target class but different bias classes. Therefore, the feature extractor $f$ learns to extract the target features while discarding the bias features.
>
> ---------
> **[Q5-1. Moving a graphical model in Appendix A to the main paper]**
>
> Thank you for the suggestion. We will move the graphical model in Appendix A to Section 3.3 to help the reader understand Equation 5 and 6. We will reflect it in the final version.
>
> ---------
> **[ Q5-2. Better notation ]**
>
> Thank you for the suggestion. We will use different notations for each distribution for better readability.
>
>
> ---------
> [a] Zemel et al., Learning fair representations, 2013
>
> [b] Madras et al., Learning adversarially fair and transferable representations, 2018
>
> [c] Creager et al., Flexibly fair representation learning by disentanglement, 2019
>
> [d] Bahng et al., Learning de-biased representations with biased representations, 2020
>
> [e] Nam et al., Learning from failure: Training debiased classifier from biased classifier, 2020
>
> [f] Tartaglione et al., EnD: Entangling and disentangling deep representations for bias correction, 2021
>
> [g] Arjovsky et al., Invariant risk minimization, 2020
>
> [h] Ganin et al., Domain-adversarial training of neural networks, 2015
>
> [i] Kim et al., Learning not to learn: Training deep neural networks with biased data, 2019
>
> [j] Oord et al., Representation learning with contrastive predictive coding, 2018

---

### Official Review · Reviewer_bmYk · 2021-07-24

**Rating:** 6
**Confidence:** 4

**Summary:**

The authors propose a Bias-Contrastive (BiasCon) loss, Soften Bias-Contrastive (SoftCon) loss and Bias-Balanced (BiasBal) regression to mitigate model biases.

**Limitations And Societal Impact:**

yes

**Main Review:**

The motivation is clear and the problem being solved is important.  The contribution seems iterative but results are strong.  I have some concerns/questions on some of the details and choices made.  Hopefully the authors can clear them up.

BiasCon pulls samples with same class but different biases closer and other pairs apart.  Why would it be preferable to pull samples with the same class apart regardless of bias?  It seems more intuitive to pull apart samples with same bias and pull together samples with same class which would also utilize same class samples without shared bias as well as different class samples with shared bias.  It would also remove the issue of limited positive pairs in a batch.  Can the authors comment on this?

Cross entropy (CE) should be pulling together samples with the same class which when combined with BiasCon in Equation 3 should balance the issue of the model learning to separate the representations of same class samples, but it seems that biasCon here would reduce the effect of the CE loss.

In the case when the dataset has limited examples of same class pairs without shared biases, which may be the case in many biased datasets, couldn't oversampling these datapoints lead to other learned biases specific to the limited data.  Essentially the model may become overtrained on these limited datapoints.

It is confusing to refer to P_u as the unbiased test data distribution as it is defined using only the training data.

Equation 8 is summing logits with probabilities.  Logits can be large, small and possibly negative depending on the model.  How is this taken into account here so that the combination with p(y|b) has an influential contribution?

Could the bias-capturing model also be used to weakly label positive bias pairs in biasCon?  More info on the bias-capturing model would be helpful.

Without these labels how can we trust that the representations of the bias-capturing model are clustered according to biases?  Many close representations could be simply datapoints of a similar class.  How is it ensured that the bias-capturing model representation is highly sensitive to bias?  In many applications it is not as simple as reducing the size of the receptive field such as the case with gender or racial biases in data.

**Time Spent Reviewing:**

2 hours

---

> ### Author Response · Authors · 2021-08-09
> **Response to Reviewer bmYk**
>
> We sincerely thank the reviewer for the constructive feedback.
> We tried our best to address the reviewer’s concerns and questions. Hope this response answers the reviewer’s questions.
> ---------
> **[Q1. Design choice of the BiasCon loss]**
>
> The fundamental issue of learning with bias is that learning through the standard cross-entropy (CE) loss learns bias faster than signal. Therefore, in designing our BiasCon loss, we focus more on preventing the classifier from using bias information rather than grouping the same class together (The latter, we believe, will be handled well by CE loss). From this point of view, we focus more on cases belonging to the same class but with different biases. Compared to considering all instances belonging to the same class regardless of bias, this approach can be seen as a focus on harder positives. Indeed many recent studies have shown that considering hard positives [a,b] as well as hard negatives [c,d] can be advantageous in contrastive learning.
>
> Nevertheless, some variants in designing contrastive loss can be devised here, including the one pointed out by the reviewer, and we also think it is important to empirically compare their performance. Toward this, we conducted additional experiments with the following two variants on the BiasedMNIST dataset:
>
> Variant 1: positive pair - same class / negative pair - different class but same bias
>
> Variant 2: positive pair - same class / negative pair - different class regardless of bias (same as SupCon loss, but trained in joint with CE loss)
>
> ||0.999|0.997|0.995|0.99|
> |:----:|:----:|:----:|:----:|:----:|
> |Variant 1|23.0|67.7|75.9|89.1  |
> |Variant 2|52.5|75.0|84.4|90.8|
> |BiasCon|92.0|96.8|97.3|98.0|
>
> We observe that using the same class but different bias samples as a positive pair is better at preventing the model from falling into shortcuts using bias features. We think the failure of Variant 1 is due to scarce negative pairs in a highly correlated case.
>
> We think this experiment is crucial to justify the design choice of our BiasCon loss. Thank you again for the feedback.
>
> ---------
> **[Q2. On the effect of the BiasCon loss to the cross-entropy (CE) loss]**
>
> The model trained only with the CE loss strongly prefers to use the bias features, making the same class and same bias sample pairs closer. Therefore, it is indeed necessary to reduce the effect of CE loss, and we devised BiasCon loss as a kind of regularizer in this sense that can give a penalty to the model if such pairs get too close than the same class and different bias sample pairs.
>
> For comparison, we tested additional variant as below on BiasedMNIST dataset, which removes same class samples in negative pairs:
>
> Variant 3: positive pair - same class but different bias / negative pair - different class
>
> ||0.999|0.997|0.995|0.99|
> |:----:|:----:|:----:|:----:|:----:|
> |Variant 3|83.0|87.7|88.5|89.4|
> |BiasCon|92.0|96.8|97.3|98.0|
>
> This empirically shows that giving a penalty to samples with the same class and same bias samples is important for learning unbiased representation.
>
> ---------
> **[Q3. On the memorization issue of the oversampling strategy]**
>
> We agree that the oversampling strategy may lead to overfitting or memorization and the model may fail to generalize.
> To prevent this, we set a bound to the amount of oversampling with \gamma in Equation (4) of our paper.
> Moreover, we separated the batch sampling strategy for CE loss and our BiasCon loss and only applied the oversampling strategy for the BiasCon loss calculation.
> This reduces the possible issues of the oversampling strategy.
>
> In Table 1, we can find that the model trained with our BiasCon loss shows high unbiased accuracy even for high correlation cases.
> This generalization ability implies that our oversampling strategy does not seriously cause the memorization issue.
>
> ---------
> **[Q4. Definition of $P_u$ in Equation 5, 6]**
>
> The intent of defining $P_u$ using the training data distribution was to emphasize the situation that while the data generation process of $P_{train}(X|Y, B)$ is unchanged even in the test phase, the correlation between $P(Y|B)$ is changed to become uniform (i.e., uncorrelated).
> We will clarify and add an explicit explanation about $P_u$ in the final version.
>
> ---------
> **[Q5. Typo in Equation 8]**
>
> We apologize for the typo in Equation 8.
> We found the typo during the submission of the supplementary material and added the correct equation in Appendix A.1.
> The correct version of Equation 8 (BiasBal loss) is $L(h(x), y, b) = -\log \frac{\exp{(\eta_y + \log p_{train}(y|b))}}{\sum_{y'} \exp{(\eta_{y'} + \log  p_{train}(y'|b))}}$.
> We will fix this in the final version. Thank you for the check.
>
> ---------
> **[Q6. Using the bias-capturing model as a weak positive bias pair labeler]**
>
> The bias-capturing model is a model that mostly uses the bias features to make a biased prediction and the idea is borrowed from the previous literature [e, f]
> The purpose of the bias-capturing model is to replace the bias label used in BiasCon.
> Regarding the use of the bias-capturing model proposed by the reviewer, in fact, we have already used the distance in the feature space of the bias-capturing model as a weak label of positive pairs in BiasCon (our SoftCon loss (Equation 10)). If you are suggesting another usage, please let us know then we will be happy to make further experiments.
>
> ---------
> **[Q7-1. On the dependency of the SoftCon loss to the bias-capturing model]**
>
> We agree that our SoftCon loss can be highly affected by the performance of the bias-capturing model. Thus, in Appendix D.1 and Table A12, we provided an experimental study of the robustness of our SoftCon loss to the failure of the bias-capturing model with the BiasedMNIST dataset. We simulated deteriorated bias-capturing models by constructing a model with a large kernel size and training on less correlated datasets. In this way, the models become less biased toward background color and can be used to simulate a situation when the bias-capturing model is unreliable. For convenience, we copied Table A12 in Appendix D.1.
>
> |Corr|ReBias[f]|LfF[g]|SoftCon|F1|F2|F3|F4|
> |:----:|:----:|:----:|:----:|:----:|:----:|:----:|:----:|
> |0.999|22.7|12.1|86.1|69.9|52.6|50.0|43.0|
> |0.997|64.2|50.3|94.7|91.8|83.1|76.9|65.2|
> |0.995|76.0|89.8|97.0|96.1|89.9|85.2|80.3|
> |0.99|88.1|94.2|97.8|97.5|95.6|93.5|90.2|
>
> Here, F denotes the failure level of the bias-capturing model. We measure the failure of the bias-capturing model with biased accuracy. (i.e., the accuracy of the model on samples with correlated biases.) Specifically, the biased accuracies for each level are 89%, 50%, 34%, and 21%, respectively, which is much lower than the 99% biased accuracy of the original bias-capturing model used for the SoftCon loss.
>
> It can be seen that our model performance degrades with the performance of the bias-capturing model, but nevertheless, it still works better than the previous state-of-the-art.
> This implies that even in cases where the bias-capturing model severely fails to capture the bias features and gives noisy weak labels for positive pairs, the SoftCon loss can still benefit from it.
>
> ---------
> **[Q7-2. When designing the bias-capturing model is infeasible]**
>
> We agree that designing a bias-capturing model may be difficult when the bias is in the form of sensitive attributes such as race or gender.
> In such cases, challenging not only with our method but with all existing methods [e, f], we may still use the property that bias features are learned much faster than the target features, as in [g]. Specifically, we can train a bias-capturing model by emphasizing samples with small loss using generalized cross-entropy (GCE) loss in the early stage of the training. This way, without explicit information about the bias, the model will be more biased toward such attributes. In order to check the feasibility of this method, we made some preliminary experiments on CelebA and UTKFace datasets. For fast comparison, results of vanilla training with CE loss, EnD[h], and BiasCon are directly copied from Table 2 and 3 of our paper. Please note that EnD and BiasCon losses are not directly comparable since they use bias labels.
>
> **CelebA**
>
> |Task|Acc. Type|SoftCon|Vanilla|EnD[h]|BiasCon|
> |:----:|:----:|:----:|:----:|:----:|:----:|
> |Blonde|Unbiased|84.1|82.6|88.0|89.5|
> ||Bias-conflict|74.4|66.4|78.7|84.6|
> |Makeup|Unbiased|77.4|76.5|77.8|79.8|
> ||Bias-conflict|61.0|57.1|62.0|65.2|
>
> **UTKFace**
>
> |Task|Acc. Type|SoftCon|Vanilla|EnD[h]|BiasCon|
> |:----:|:----:|:----:|:----:|:----:|:----:|
> |Race|Unbiased|87.0|87.2|88.1| 90.4|
> ||Bias-conflict|80.2|79.1|81.4|88.9|
> |Age|Unbiased|74.6|72.4|74.9|79.0|
> ||Bias-conflict|59.2|47.5|63.2|75.0|
>
> The results show that inducing the bias-capturing model trained with GCE loss can give noisy but meaningful labels of positive pairs with shared bias that SoftCon loss can utilize.
> We believe it can be a good future research direction to deal with this issue in more depth and find a fundamental solution.
>
> ---------
> [a] Khosla et al., Supervised Contrastive Learning, 2020
>
> [b] Zhang et al., Self-supervised representation learning via adaptive hard-positive mining, 2021
>
> [c] Kalantidis et al., Hard Negative Mixing for Contrastive Learning, 2020
>
> [d] Xuan et al., Hard negative examples are hard, but useful, 2020
>
> [e] Clark et al., Don’t take the easy way out: Ensemble based methods for avoiding known dataset biases, 2019
>
> [f] Bahng et al., Learning De-biased Representations with Biased Representations, 2020
>
> [g] Nam et al., Learning from failure:Training debiased classifier from biased classifier, 2020
>
> [h] Tartaglione et al., End: Entangling and disentangling deep representations for bias correction, 2021

---

> > ### Comment · Reviewer_bmYk · 2021-08-25
> > **Clarification on Q1/Q2**
> >
> > Thank you for the detailed response.  For the approach I mentioned in my review that felt more intuitive I was suggesting using a loss with two weighted objectives which each have separate positive and negative pairs.  The first objective would be pretty standard and have positive pairs be from the same class and the negatives be from different classes.  The second objective, which can be thought of as a regularizer, would have positive pairs that have different biases, regardless of class, and negative pairs having the same bias, again regardless of class.  This is different than variant 1 and 2 that was shown in your response and more directly addresses your point in your Q2 response on the limitations of CE in this context while still being different than biasCon.  I understand that not all approaches can be run, but I was wondering on thoughts regarding this.

---

> > > ### Author Response · Authors · 2021-08-26
> > > **Further Response to Reviewer bmYk**
> > >
> > > Thank you for the further suggestion.
> > >
> > > ---------
> > > **[Design choice of the BiasCon loss (continued discussion of Q1 and Q2)]**
> > >
> > > Based on the reviewer’s suggestion, we could think of a variant of our BiasCon loss that has two objectives each of which has separate positive and negative pairs.
> > >
> > > **Variant 4**
> > >
> > > Objective 1: Contrastive loss with positive pair - same class / negative pair - diff class
> > >
> > > Objective 2: Contrastive loss with positive pair - diff bias / negative pair - same bias
> > >
> > > For comparison, we remind the reviewer of our BiasCon loss again.
> > >
> > > **BiasCon**
> > >
> > > Objective: Contrastive loss with positive pair - same class but different bias / negative pair - others
> > >
> > > Contrastive loss regularizes the model so that features of negative sample pairs are no closer than the features of positive sample pairs.
> > > In this sense, Objective 1 of Variant 4 prefers the same class samples to be closer than the different class samples, similar to the cross-entropy (CE) loss.
> > > However, Objective 2 of Variant 4 regularizes the model in a way that features of the same bias samples (which are mostly the same target class samples) are no closer than the features of the different bias samples (which are mostly the different target class samples).
> > > For an extreme example, even though the unbiased model successfully learned the target features, the loss of Objective 2 will still be high as the features of the same target class samples with the same bias (which are the majority case since we are tackling the dataset where the signal and bias are correlated) are closer than the features of the different target class samples with different biases.
> > >
> > > In contrast, our BiasCon loss only encourages the model not to map the same class and same bias samples especially closer than the same class and different bias samples, and it does not penalize even when the same target class samples are closer than the different target class samples.
> > >
> > > We will do our best for the rest of the discussion period to see if our expectation is true through experiments. We ask the reviewer to wait for our next reply containing the experimental results.

---

> > > > ### Comment · Area_Chair_qDz6 · 2021-08-31
> > > > **Discussion period coming to an end**
> > > >
> > > > Dear Authors,
> > > >
> > > > Thanks for the detailed back-and-forth with the reviewers.
> > > >
> > > > The discussion period is coming to an end on September 2nd. If you plan to provide further clarifications or additional results, please do so before the deadline.
> > > >
> > > > Regards,
> > > >
> > > > AC

---

> > > > > ### Author Response · Authors · 2021-09-01
> > > > > **Experiment Results of the Further Response Added**
> > > > >
> > > > > Thank you for the notice.
> > > > >
> > > > > We’ve posted the experiment results about the variants explained in the Further Response to Reviewer bmYk.

---

> > > ### Author Response · Authors · 2021-09-01
> > > **Experiment Results of the Further Response**
> > >
> > > We’ve evaluated the performance of Variant 4 on the BiasedMNIST dataset.
> > >
> > > **Variant 4**
> > >
> > > Objective 1: Contrastive loss with positive pair - same class / negative pair - diff class
> > >
> > > Objective 2: Contrastive loss with positive pair - diff bias / negative pair - same bias
> > >
> > > Variant 4 replaces the CE loss with Objective 1, and the contrastive loss is only applied to the penultimate layer.
> > > Therefore, following the SupCon loss [a], we expect Variant 4 requires two-stage training that uses the loss of Variant 4 to train the feature representation and train the classifier head with CE loss upon the frozen network.
> > > This implies that Variant 4 gives an additional burden of two-stage training.
> > >
> > > For a fair comparison, we used the same oversampling strategy used for our BiasCon loss.
> > >
> > > ||0.999|0.997|0.995|0.99|
> > > |:----:|:----:|:----:|:----:|:----:|
> > > |Variant 4|66.6|85.1|90.2|93.4|
> > > |BiasCon|92.0|96.8|97.3|98.0|
> > >
> > > The results show that Variant 4 shows worse performance than our BiasCon loss.
> > > As explained in the previous answer, we expect that Objective 2 of Variant 4 is the culprit of the performance degradation.
> > > Objective 2 promotes the feature of the same bias samples to be no closer than the different bias samples.
> > > However, as the correlation between target and bias gets greater, the majority of the same bias samples have the same target and vice versa.
> > > In other words, we can expect that Objective 2 promotes features of the same target and same bias samples (which are the majority case) to be no closer than the features of different target and different bias samples.
> > > However, our BiasCon loss promotes the feature of the same target and same bias samples to be no closer than the same target and different bias samples.
> > > We expect that this critical difference makes Variant 4 worse than our BiasCon loss.
> > >
> > > We will add experiments on the suggested variants to justify the design choice of our BiasCon loss in the final version.
> > >
> > > Thank you again for the constructive feedback.
> > >
> > > ---------
> > > [a] Khosla et al., Supervised Contrastive Learning, 2020

---

> > > > ### Comment · Reviewer_bmYk · 2021-09-01
> > > > **Acknowledgment of the response**
> > > >
> > > > Thank you for the detailed responses and the additional experiments.  This satisfies my main concerns and I will recommend acceptance.

---

### Decision · Program_Chairs · 2021-09-27

**Decision:**

Accept (Poster)

**Comment:**

The paper proposes new loss functions for training on datasets with feature bias. The reviewers find the proposals to be novel and the experiment results to be strong, and recommend acceptance. There were some concerns raised about the rationale behind some of the design choices, and missing comparisons to some baselines. The authors have satisfactorily addressed these concerns over multiple back-and-forth discussions with the reviewers, and have provided additional supporting experimental results. We trust that the authors will include the additional results in the final version of the paper.